# Proteolytically released Lasso/teneurin-2 induces axonal attraction by interacting with latrophilin-1 on axonal growth cones

**Nickolai V Vysokov**[1,2,3,4], **John-Paul Silva**[2,5], **Vera G Lelianova**[1,2], **Jason Suckling**[2,6], **John Cassidy**[2,7], **Jennifer K Blackburn**[1,8], **Natalia Yankova**[2,9], **Mustafa BA Djamgoz**[2], **Serguei V Kozlov**[10], **Alexander G Tonevitsky**[11,12], **Yuri A Ushkaryov**[1,2]*

[1]School of Pharmacy, University of Kent, Chatham, United Kingdom; [2]Department of Life Sciences, Imperial College London, London, United Kingdom; [3]Wolfson Centre for Age Related Diseases, King's College London, London, United Kingdom; [4]BrainPatch Ltd, London, United Kingdom; [5]Department of Bioanalytical Sciences, Non-clinical development, UCB-Pharma, Berkshire, United Kingdom; [6]Thomsons Online Benefits, London, United Kingdom; [7]Arix Bioscience, London, United Kingdom; [8]Division of Molecular Psychiatry, Yale University School of Medicine, New Haven, United States; [9]Institute of Psychiatry, Psychology & Neuroscience, Maurice Wohl Clinical Neuroscience Institute, Department of Basic and Clinical Neuroscience, King's College London, London, United Kingdom; [10]Center for Advanced Preclinical Research, National Cancer Institute, Frederick, United States; [11]Higher School of Economics, Moscow, Russia; [12]Scientific Research Centre Bioclinicum, Moscow, Russia

*For correspondence:
Correspondence: y.ushkaryov@
kent.ac.uk

**Abstract** A presynaptic adhesion G-protein-coupled receptor, latrophilin-1, and a postsynaptic transmembrane protein, Lasso/teneurin-2, are implicated in trans-synaptic interaction that contributes to synapse formation. Surprisingly, during neuronal development, a substantial proportion of Lasso is released into the intercellular space by regulated proteolysis, potentially precluding its function in synaptogenesis. We found that released Lasso binds to cell-surface latrophilin-1 on axonal growth cones. Using microfluidic devices to create stable gradients of soluble Lasso, we show that it induces axonal attraction, without increasing neurite outgrowth. Using latrophilin-1 knockout in mice, we demonstrate that latrophilin-1 is required for this effect. After binding latrophilin-1, Lasso causes downstream signaling, which leads to an increase in cytosolic calcium and enhanced exocytosis, processes that are known to mediate growth cone steering. These findings reveal a novel mechanism of axonal pathfinding, whereby latrophilin-1 and Lasso mediate both short-range interaction that supports synaptogenesis, and long-range signaling that induces axonal attraction.
DOI: https://doi.org/10.7554/eLife.37935.001

## Introduction

Correct wiring of the nervous system critically depends on both long-range diffusible cues and short-range contact-mediated factors which can be attractive or repulsive (*Chen and Cheng, 2009*). However, the relatively small repertoire of key molecules known to be involved in axon guidance or trans-synaptic adhesion cannot fully explain the complexity and specificity of synaptic connections. Indeed, new interacting partners and signal-modulating ligands are now being found for many well-

**eLife digest** The brain is a complex mesh of interconnected neurons, with each cell making tens, hundreds, or even thousands of connections. These links can stretch over long distances, and establishing them correctly during development is essential. Developing neurons send out long and thin structures, called axons, to reach distant cells. To guide these growing axons, neurons release molecules that work as traffic signals: some attract axons whilst others repel them, helping the burgeoning structures to twist and turn along their travel paths.

When an axon reaches its target cell, the two cells join to each other by forming a structure called a synapse. To make the connection, surface proteins on the axon latch onto matching proteins on the target cell, zipping up the synapse. There are many different types of synapses in the brain, but we only know a few of the surface molecules involved in their creation – not enough to explain synaptic variety.

Two of these surface proteins are latrophilin-1, which is produced by the growing axon, and Lasso, which sits on the membrane of the target cell. The two proteins interact strongly, anchoring the axon to the target cell and allowing the synapse to form. However, a previous recent discovery by Vysokov et al. has revealed that an enzyme can also cut Lasso from the membrane of the target cell. The 'free' protein can still interact with latrophilin-1, but as it is shed by the target cell, it can no longer serve as an anchor for the synapse. Could it be that free Lasso acts as a traffic signal instead?

Here, Vysokov et al. tried to answer this by growing neurons from a part of the brain called the hippocampus in a special labyrinth dish. When free Lasso was gradually introduced in the culture through microscopic channels, it interacted with latrophilin-1 on the surface of the axons. This triggered internal changes that led the axons to add more membrane where they had sensed Lasso, making them grow towards the source of the signal.

The results demonstrate that a target cell can both carry and release Lasso, using this duplicitous protein to help attract growing axons as well as anchor them. The work by Vysokov et al. contributes to our knowledge of how neurons normally connect, which could shed light on how this process can go wrong. This may be relevant to understand conditions such as schizophrenia and ADHD, where patients' brains often show incorrect wiring.
DOI: https://doi.org/10.7554/eLife.37935.002

established guidance factors (*Karaulanov et al., 2009*; *Leyva-Díaz et al., 2014*; *Söllner and Wright, 2009*). Furthermore, our novel findings demonstrate that at least one receptor pair can both mediate cell contacts and, unexpectedly, also act as a long-range signaling factor and its receptor.

This trans-synaptic receptor pair consists of presynaptic latrophilin-1 (LPHN1) and postsynaptic Lasso (*Silva et al., 2011*). LPHN1 (also known as ADGRL1 for *Adhesion G-protein-coupled Receptor, Latrophilin subfamily 1* [*Hamann et al., 2015*]) is a cell-surface receptor that is expressed by all central neurons (*Davletov et al., 1998*; *Ichtchenko et al., 1999*; *Matsushita et al., 1999*; *Sugita et al., 1998*). An array of data indicates that LPHN1 is localized on axons, axonal growth cones and nerve terminals (*Silva et al., 2011*). Activation of LPHN1 by its agonist, mutant latrotoxin (LTX$^{N4C}$), stimulates vesicular exocytosis (*Ashton et al., 2001*; *Lajus et al., 2006*; *Lelyanova et al., 2009*; *Silva et al., 2009*; *Tobaben et al., 2002*; *Volynski et al., 2003*; *Deák et al., 2009*). LPHN1 knockout (KO) in mice leads to abnormal rates of embryonic lethality and psychotic phenotypes (*Tobaben et al., 2002*), indicating the importance of LPHN1 in early development and in cognitive functions in adulthood.

The second member of this receptor pair, Lasso, is a representative of teneurins (TENs), large single-pass transmembrane proteins (*Baumgartner et al., 1994*; *Levine et al., 1994*). Lasso is the splice variant of TEN2 (TEN2-SS) (*Figure 1A*) that specifically binds LPHN1 in cell adhesion experiments (*Li et al., 2018*). Given also that only Lasso is isolated by affinity chromatography on LPHN1 (*Silva et al., 2011*), we will refer here to TEN2 that is able to bind LPHN1 as Lasso. All TENs possess a large C-terminal extracellular domain (ECD) containing a series of epidermal growth factor (EGF)-like repeats and other repeat domains (*Figure 1A*). Inter-chain disulfide bridges mediate TEN homo-dimerization (*Figure 1B*, left) (*Feng et al., 2002*; *Vysokov et al., 2016*). Similar to Notch, during the intracellular processing of TENs, their ECDs are constitutively cleaved by furin at site 1 (*Figure 1A,B*,

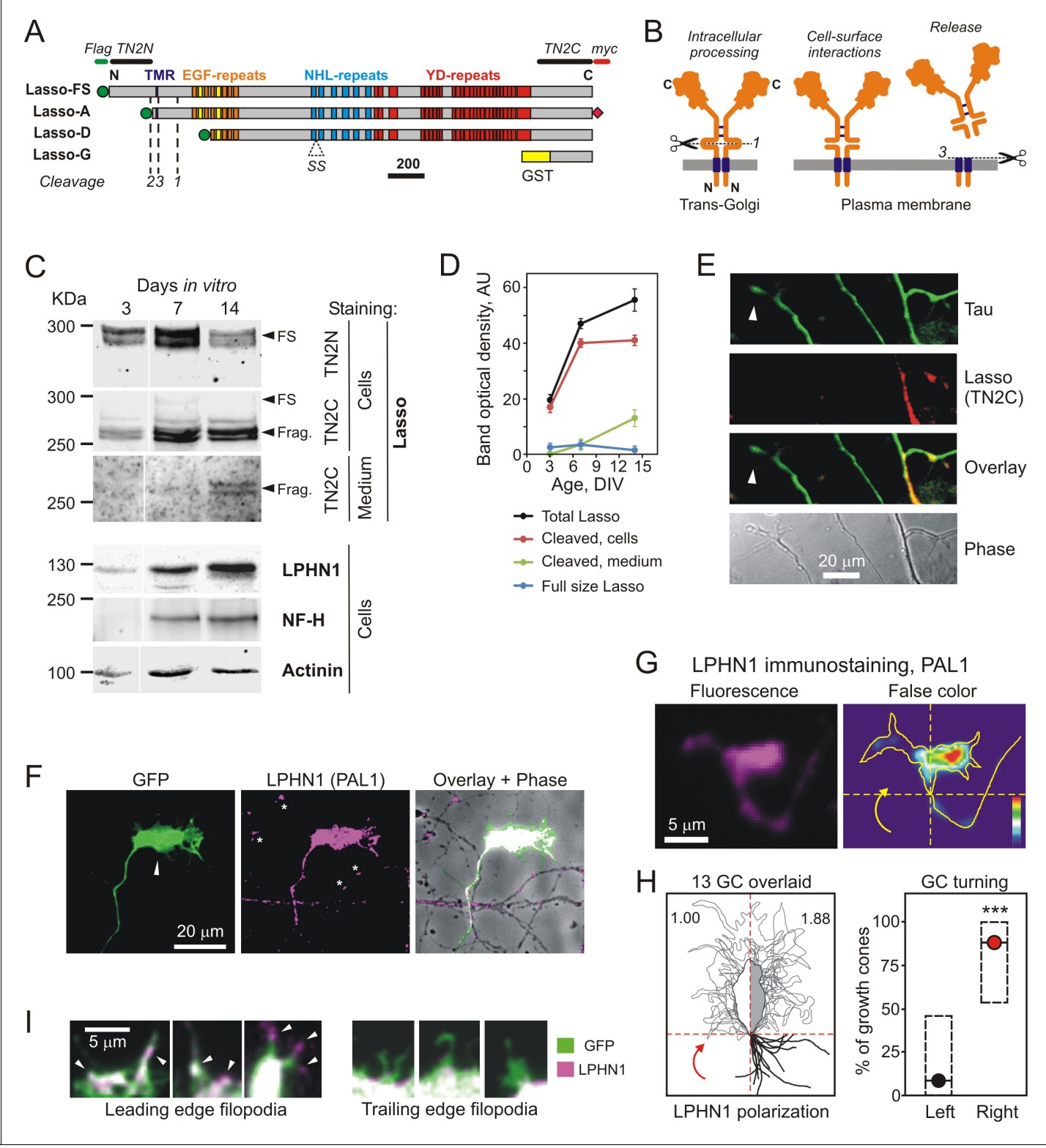

**Figure 1.** Lasso is cleaved and released into the medium during neuronal development. (**A**) Recombinant Lasso constructs used in this work (FS, full size). The three proteolytic cleavage sites and the SS splice site are indicated. The antibody recognition sites/epitopes are shown by bars above the structure. Scale bar, 200 amino acids. (**B**) Intracellular processing and release of TENs. Left, TEN2 is constitutively cleaved in the trans-Golgi vesicles by furin at site 1. Middle, when delivered to the cell surface, the ECD remains tethered to the membrane and functions as a cell-surface receptor. Right, regulated cleavage at site 3 releases the ECD into the medium. (**C**) Expression of Lasso and release of its ECD fragment in hippocampal neurons in

*Figure 1 continued on next page*

*Figure 1 continued*

culture. Rat hippocampal neurons were cultured for 3, 7 and 14 days, and proportionate amounts of the conditioned media and cell lysates were separated by SDS-PAGE. A Western blot (representative of three independent experiments, which all gave similar results) was stained for Lasso, LPHN1, neurofilament-H (NF-H), and actinin. The doublet bands corresponding to splice variants of full-size Lasso (FS) and the fragment of ECD (Frag.) cleaved at site 1 are indicated by arrowheads. (D) Quantification of Western blots (as in C), using Lasso C-terminus staining data. (E) Axonal growth cones (white arrowheads) do not express Lasso/teneurin-2. Neurons in a 9 DIV hippocampal culture were permeabilized and stained for the axonal protein Tau (green) and Lasso (TN2C, red) (representative image from $n$ = 5 experiments). (F) A detailed study of growth cones. Hippocampal neurons were transfected with a vector encoding GFP, then, after 14 DIV, stained for LPHN1 (PAL1 and Alexa 647-conjugated secondary antibody, magenta), and axonal growth cones were visualized by GFP fluorescence (green). (G, H) Correlation of LPHN1 polarization within a growth cone with its recent travel trajectory. G left, a fluorescent image of a growth cone stained for LPHN1 (magenta). G right, the same image in false color (contour based on GFP staining), demonstrating LPHN1 polarization on the right side. H left, the contours of 13 roughly symmetrical growth cones and their preceding axons were aligned to locate the stronger LPHN1 staining on the right. Note, that all axons approach growth cones from the right low quadrant. H right, the proportion of right- and left-turning growth cones plotted with Jeffreys 99.73% confidence intervals for a binomial parameter; \*\*\*, p<0.001; $n$ = 13. (I). LPHN1 is found within filopodia and lamellipodia on the leading edge (left, arrowheads), but not on the trailing edge (right) of a growth cone. Green, GFP fluorescence; magenta, PAL1 staining for LPHN1.

DOI: https://doi.org/10.7554/eLife.37935.003

The following source data and figure supplements are available for figure 1:

**Source data 1.** Source data for *Figure 1*, Panels D and H.
DOI: https://doi.org/10.7554/eLife.37935.004
**Figure supplement 1.** Lasso is expressed on dendrites and LPHN1 on axonal growth cones in developing neurons.
DOI: https://doi.org/10.7554/eLife.37935.005
**Figure supplement 1—source data 1.** Source data for *Figure 1—figure supplement 1*, Panels A and G.
DOI: https://doi.org/10.7554/eLife.37935.006

left) (*Rubin et al., 1999*; *Tucker and Chiquet-Ehrismann, 2006*; *Vysokov et al., 2016*). However, the cleaved ECD remains tightly tethered to the cell surface due to its strong interaction with the transmembrane fragment (*Figure 1B*, middle) (*Vysokov et al., 2016*).

TENs have been implicated in promoting axon guidance and neurite outgrowth (*Minet et al., 1999*; *Rubin et al., 1999*; *Antinucci et al., 2013*; *Leamey et al., 2007*; *Young et al., 2013*; *Hor et al., 2015*). For example, different TENs can mediate neuronal cell adhesion (*Boucard et al., 2014*; *Rubin et al., 2002*; *Silva et al., 2011*). TEN2 and TEN4, which are present on dendritic growth cones and developing filopodia, may be responsible for dendritic spine formation (*Rubin et al., 1999*; *Suzuki et al., 2014*), while substrate-attached TEN1 supports neurite growth (*Minet et al., 1999*). However, a mechanistic insight into the role of TENs in axonal growth is still lacking.

One possibility is that TENs, as *bona fide* cell-surface receptors, could bind other cell-surface molecules and thus mediate axonal pathfinding. TENs can form homophilic complexes (*Rubin et al., 2002*; *Beckmann et al., 2013*). However, TENs failed to mediate homophilic cell adhesion in direct experiments (*Boucard et al., 2014*; *Li et al., 2018*). In addition, homophilic interactions of a recombinant soluble TEN2 ECD with the cell-surface TEN2 inhibited (rather than promoted) neurite outgrowth (*Beckmann et al., 2013*; *Young et al., 2013*). By contrast, heterophilic interactions of TENs can promote synapse formation (*Mosca et al., 2012*; *Silva et al., 2011*). More specifically, heterophilic interaction between Lasso and LPHN1, its strongest ligand (*Silva et al., 2011*; *Boucard et al., 2014*), consistently mediates cell adhesion (*Silva et al., 2011*; *Boucard et al., 2014*; *Li et al., 2018*) and is thought to facilitate synapse formation (*Silva et al., 2011*).

However, our surprising finding (*Vysokov et al., 2016*) that Lasso/TEN2 is partially released from the cell surface by regulated proteolysis (at site 3; *Figure 1B*, right) was inconsistent with a solely cell-surface function of Lasso. On the other hand, we found that the released Lasso fragment retained its ability to bind cell-surface LPHN1 with high affinity and induce intracellular signaling (*Silva et al., 2011*; *Vysokov et al., 2016*). Thus, it was possible that the released, soluble ECD of Lasso/TEN2 could act as a diffusible (attractive or repulsive) factor and mediate some of the TEN2 functions in neurite pathfinding described above. Therefore, we hypothesized that the binding of soluble Lasso to LPHN1 on distant neurites could trigger important changes in their growth.

Here, we test this hypothesis using cultured hippocampal neurons. First, we show that developing neurons release a substantial proportion of Lasso ECD into the medium, while LPHN1 is concentrated on the leading edge of axonal growth cones. We then use microfluidic chambers to

demonstrate that a spatio-temporal gradient of soluble Lasso attracts neuronal axons, but not dendrites, and that this process involves LPHN1 that is present on axonal growth cones. Using model cells expressing functional LPHN1, and mouse neuromuscular preparations, we also show that LPHN1 activation by soluble Lasso causes intracellular $Ca^{2+}$ signaling, which leads to increased exocytosis. This suggests a plausible cellular mechanism causing axons to turn in the direction of a gradient of soluble Lasso. Moreover, the LPHN1-Lasso pair illustrates a novel principle of chemical guidance whereby cell-surface receptors engage not only in short-range interactions, but also in long-range signaling, which can further contribute to the formation of complex neuronal networks.

## Results

### Neurons partially cleave and release lasso

We previously showed in model cell lines and in adult brain that Lasso is cleaved at several sites (sites 1, 2, three in *Figure 1A,B*) and is released into the extracellular environment in a regulated manner (*Vysokov et al., 2016*). To test whether Lasso undergoes the same processing and release during neuronal development, we followed Lasso expression at different stages of neuron maturation in hippocampal cell cultures (*Kaech and Banker, 2006*). Soon after plating, embryonic (E18) rat hippocampal neurons produced Lasso, which was detectable at 3 days in vitro (DIV) (*Figure 1C,D*). A large proportion of Lasso (~90%) was constitutively cleaved at site 1 during neuronal development in vitro (*Figure 1—figure supplement 1A*). Increasing amounts of cleaved fragment also appeared in the medium at 7 and 14 DIV (*Figure 1D* and *Figure 1—figure supplement 1A*, green), indicating a slow cleavage at site 3. Thus, Lasso is fully cleaved at site 1 and partially released by regulated cleavage at site 3 not only in transfected immortalized cells, but also in developing neurons and in the postnatal rat brain (*Vysokov et al., 2016*).

We also examined the neuronal structures that could release soluble Lasso ECD. We found that large amounts of Lasso were present on dendrites and dendritic growth cones (*Figure 1—figure supplement 1B*), while it was practically absent from axons and axonal growth cones (*Figure 1E*). Since about 80% of Lasso was not normally released (*Figure 1D*, *Figure 1—figure supplement 1A*), these data suggested that the compartments rich in Lasso, that is dendrites and dendritic growth cones, were the main source of the soluble Lasso fragment.

### LPHN1 is expressed on growth cones of developing neurons

As early as 3 DIV, the developing neurons also expressed LPHN1, the high-affinity receptor for soluble Lasso ECD, and the amounts of LPHN1 continued to increase through all time points (*Figure 1B*), in parallel with the increasing amounts of soluble Lasso (*Figure 1—figure supplement 1A*). This correlation between the soluble Lasso and cell-surface LPHN1 further supported the idea of their likely interaction during neuronal development.

Interestingly, in developing hippocampal neurons, LPHN1 was found concentrated in axons and especially in axonal growth cones, where it co-localized with synapsin (*Figure 1—figure supplement 1C, D*, arrowheads). LPHN1 was also enriched in axonal varicosities, which were identified as *en passant* synapses by immunostaining for PSD-95 (*Figure 1—figure supplement 1D*, asterisks).

We then studied the expression of LPHN1 in growth cones in more detail by transfecting hippocampal neurons with GFP, which greatly simplified the identification and tracking of axons and axonal growth cones. All GFP-labeled axonal growth cones showed a clear enrichment of endogenous LPHN1 (*Figure 1F,G,I*). Conversely, when LPHN1 expression was knocked down by shRNA (delivered together with GFP in the same bicistronic vector), it clearly disappeared from the growth cones of transfected neurons, while the growth cones of non-transfected cells were not affected (*Figure 1—figure supplement 1E*, arrow and arrowhead, respectively).

We also discovered that endogenous LPHN1 expression within axonal growth cones was polarized in relation to the cone's symmetry axis, such that one side of each growth cone contained on average $1.88 \pm 0.22$ fold more LPHN1 than the other (*Figure 1G,H*). To assess whether this LPHN1 enrichment correlated with the direction of axonal growth, we traced the growth trajectories of a number of symmetrical growth cones and compared these with the distribution of LPHN1. This analysis clearly demonstrated that LPHN1 polarization within the growth cones very strongly positively correlated with the direction of their turning (*Figure 1G,H*). Moreover, in non-symmetrical growth

cones, which had clearly started turning prior to fixation, LPHN1 expression had a bimodal distribution, being enriched not only near the 'neck' of a turning cone, but also close to its leading edge (*Figure 1—figure supplement 1F, G*). Such leading-edge enrichment also extended into fine growth cone protrusions. Thus, filopodia and lamellipodia located on the leading edge of a growth cone (*Figure 1I*, left, arrowheads) showed a much higher amount of LPHN1 than the processes on the trailing edge of the growth cone (*Figure 1I*, right).

We concluded that LPHN1 expression within growth cones correlated positively with the global directionality of growth and with the fine structures that underpin the growth cone's extension.

## Soluble Lasso binds to cell-surface LPHN1

Next, we tested the interaction between soluble Lasso and cell-surface LPHN1. For these tests we expressed a shorter, constitutively secreted construct, Lasso-D (*Figure 2A*, right) in HEK293A cells and affinity-purified it (*Figure 2B*). 100 nM Lasso-D was incubated with neuroblastoma cells stably expressing (i) LPHN1, (ii) a chimeric construct LPH-82 containing ECD from EMR-2 used as a negative control, (iii) Lasso-A, or (iv) Lasso-FS (*Figure 2A*, left). As expected, Lasso-D did not interact with LPH-82 (*Figure 2C*, panel 4). The lack of Lasso-D binding to Lasso-A and released fragment of Lasso-A binding to Lasso-FS (*Figure 2D*, panels 2, 3; *Figure 2—figure supplement 1B*) was somewhat surprising, since homophilic interactions between membrane-bound and soluble TENs were reported previously (*Bagutti et al., 2003*; *Beckmann et al., 2013*; *Hong et al., 2012*; *Rubin et al., 2002*; *Boucard et al., 2014*), but this could be due to a relatively low affinity of Lasso-Lasso interaction and relatively long washes employed in our protocol. On the other hand, and consistent with previous reports of high affinity between LPH1 and Lasso (*Silva et al., 2011*; *Boucard et al., 2014*), Lasso-D and the released fragment of Lasso-A bound strongly to cells expressing LPHN1 (*Figure 2C*, panels 2, three and *Figure 2—figure supplement 1A*).

To verify that the soluble ECD of Lasso, when proteolytically released from the cell-surface as depicted in *Figure 2A* (Lasso-A), could diffuse between individual cells and bind LPHN1 on distant cells, we co-cultured neuroblastoma cells stably expressing Lasso-A with cells stably expressing LPHN1. When co-cultured at high density, these cells formed clusters, held together by LPHN1/Lasso-A intercellular adhesion complexes (*Figure 2E*, panel 1). In more sparsely plated co-cultures, the Lasso-A fragment was released into the medium, where it diffused and bound to cells expressing LPHN1, but not to the wild type (WT) neuroblastoma cells (*Figure 2E*, panel 2, and *Figure 2—figure supplement 1C*). Interestingly, after binding Lasso, the LPHN1 staining appeared to concentrate in large patches, a pattern very different from LPHN1 distribution in control conditions (*Figure 2C*, panel 1) (see also below). These experiments suggest that (i) when Lasso is released into the medium as a result of its regulated cleavage, it retains its affinity for LPHN1 and (ii) on reaching distant LPHN1-expressing cells by diffusion, Lasso causes LPHN1 redistribution on the cell surface.

We then asked whether the soluble Lasso ECD could similarly bind to LPHN1 in neurons and, more specifically, on axonal growth cones. To control for the specificity of Lasso binding to LPHN1, this experiment was carried out on cultured hippocampal neurons from LPHN1 WT (*Adgrl1$^{+/+}$*) and LPHN1 KO (*Adgrl1$^{-/-}$*) newborn mice (P0). Also, to unequivocally distinguish between the soluble and cell-surface Lasso, we used exogenous Lasso-D, which was detected using anti-FLAG antibody. As expected, in WT mouse neurons, LPHN1 was found mostly in axonal growth cones (arrowheads) and varicosities (asterisks) (*Figure 2—figure supplement 2A*, green). The exogenous Lasso-D clearly bound to these structures (*Figure 2—figure supplement 2A*, red; C), but in general did not interact with dendrites. By contrast, the axons and growth cones of LPHN1 KO neurons did not show specific LPHN1 staining and appeared unable to bind the soluble exogenous Lasso-D (*Figure 2—figure supplement 2B,C*). These results indicated that released Lasso ECD could interact with LPHN1 on axonal growth cones.

## MAIDs as a tool to study axonal responses to chemoattractant gradients

Based on the data above, we hypothesized that the interaction of released Lasso ECD with LPHN1 on axonal growth cones could represent one of the mechanisms that underlie the previously formulated, but so far unexplained, role of TENs in axonal pathfinding and brain patterning (*Antinucci et al., 2013*; *Hor et al., 2015*; *Leamey et al., 2007*; *Young et al., 2013*). To

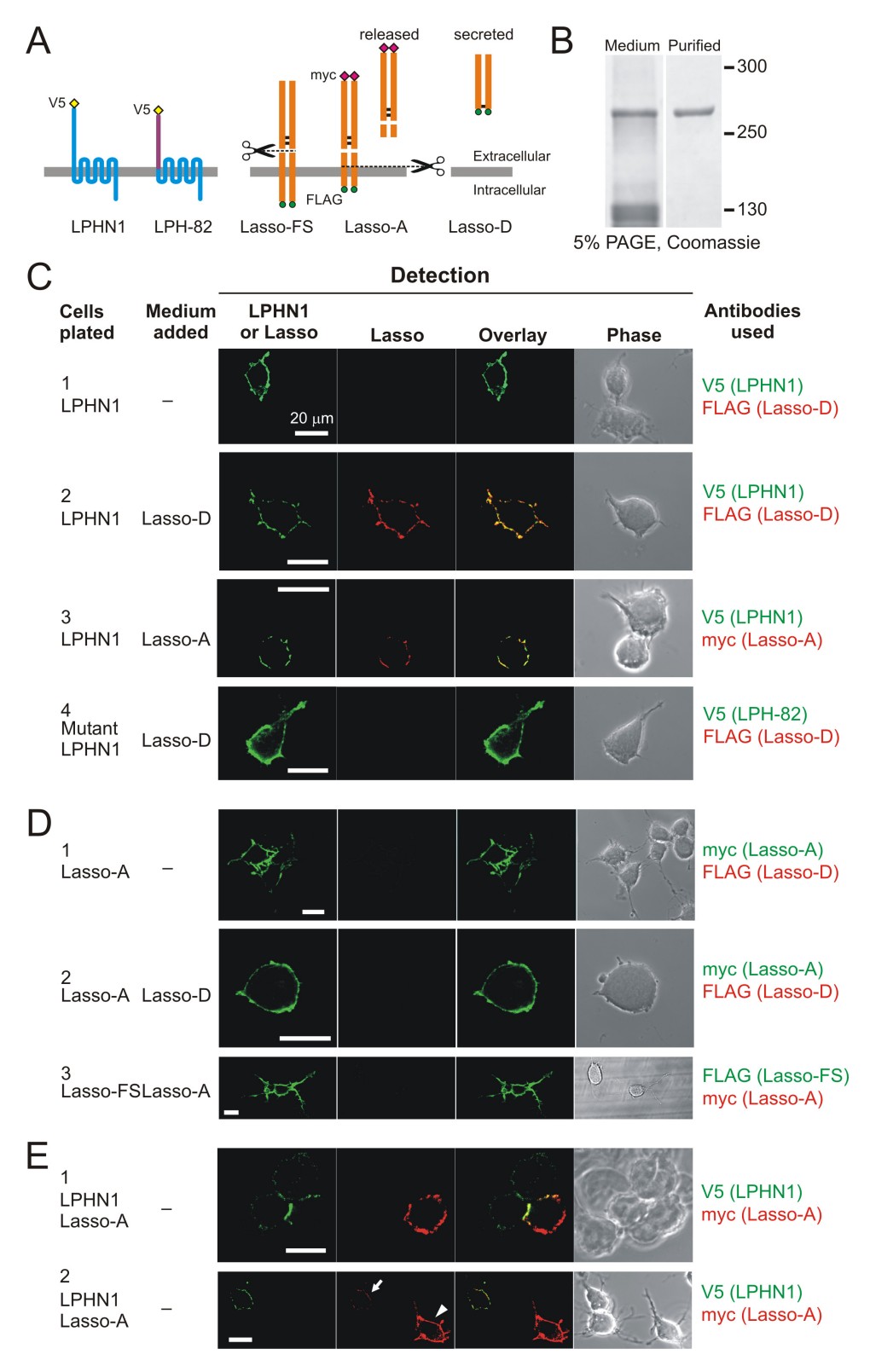

**Figure 2.** Soluble Lasso binds to LPHN1 on other cells. (**A**) A scheme of LPHN and Lasso constructs used in this experiment. LPH-82 is LPHN1 with the ECD from another adhesion G-protein-coupled receptor, EMR2, used as a negative control. (**B**) Purification of Lasso-D. Lasso-D was expressed in stably transfected HEK293 cells, then purified on a column with anti-FLAG Ab and analyzed by SDS-PAGE in a 5% gel, stained with Coomassie R250. (**C–E**) Interaction between the soluble Lasso species and NB2a cells expressing LPHN1, LPH-82, or Lasso-A. Cells expressing LPHN1 (C, panels 2, 3), but not
*Figure 2 continued on next page*

*Figure 2 continued*

Lasso-A or Lasso-FS (D) or mutant LPH-82 (C, panel 4) are able to interact with Lasso-D or Lasso-A. E, panel 1. Short-term, high-density incubation of cells expressing LPHN1 and membrane-anchored Lasso-A allows these proteins to form inter-cellular contacts. E, panel 2. After a 48 hr co-culture, a sufficient amount of Lasso-A is released into the medium, diffuses away from Lasso-A expressing cells (arrowhead) and can be detected interacting with distant LPHN1-expressing cells (arrow). Images are representative of *n* = 6–7 independent experiments.

DOI: https://doi.org/10.7554/eLife.37935.007

The following source data and figure supplements are available for figure 2:

**Figure supplement 1.** Soluble Lasso specifically binds to LPHN1-expressing cells.

DOI: https://doi.org/10.7554/eLife.37935.008

**Figure supplement 1—source data 1.** Source data for *Figure 2—figure supplement 2*, Panel C.

DOI: https://doi.org/10.7554/eLife.37935.009

**Figure supplement 2.** Soluble Lasso specifically binds to LPHN1 on axonal growth cones.

DOI: https://doi.org/10.7554/eLife.37935.010

study this effect, we developed a new method of long-term exposure of hippocampal axons to stable gradients of Lasso using 'microfluidic axon isolation devices' (MAIDs) (*Figure 3A*). The advantage of this method over conventional ligand-puffing was that the MAIDs enabled exposure of axons to long-term stable gradients of Lasso, which was critical for our assay. The device used here had two compartments, each consisting of two cylindrical wells connected by a 'corridor'; a 150 µm-thick wall that separated the two corridors had multiple parallel microchannels (2–3 µm tall and 10 µm wide) connecting the two compartments (*Figure 3A*, middle). When neurons are plated in one of the compartments (designated as the Somal Compartment), their neurites grow in all directions, but only the axons (identified by NF-H staining) readily penetrate the microchannels and cross into the empty, Axonal Compartment (*Figure 3A*, right; 3B, C). While there is a large number of dendrites in the Somal Compartment (identified by microtubule-associated protein 2, MAP-2, staining), only a few of them enter the Axonal Compartment and then terminate close to the wall (*Figure 3B,C*).

From the previously described physical characteristic of microfluidic chambers (*Zicha et al., 1991*), we predicted that a concentration gradient across the microchannels in our devices could be established over time. This was modelled by adding TRITC-conjugated BSA to one compartment and visualizing the dye in the microchannels (*Figure 3D*). We found that a gradient was formed within the first 24 hr and remained stable over several days (*Figure 3D,E*).

To test the functionality of the MAIDs for studying axonal guidance, we employed brain-derived neurotrophic factor (BDNF) known to act as an axonal chemoattractant (*Li et al., 2005*). Rat hippocampal neurons were plated into the Somal Compartment, and at 3 DIV, when axons normally start entering microchannels, BDNF was added to the Axonal Compartment (PBS was added to control cultures) (*Figure 3F*). After a further 5 DIV, we observed a 2.2-fold higher number of axons crossing into the Axonal Compartment in the presence of BDNF compared with the control (*Figure 3G,H*). This effect was statistically significant (*Figure 3H*). This proof-of-concept experiment confirmed that MAIDs could be used to study the long-term effects of chemoattractant gradients on axonal migration.

## A gradient of soluble lasso induces axonal attraction

We then used this methodology to study the reaction of LPHN1-expressing neuronal growth cones to a gradient of soluble released Lasso. Lasso-D was added to the Axonal Compartment (*Figure 4A*), and the integrity of Lasso during the experiment was verified by Western blotting (*Figure 4B*). Quantification of axons in Axonal Compartments by NF-H immunofluorescence (*Figure 4C,D*) revealed a statistically significant 1.5-fold increase in axonal growth induced by Lasso-D. Thus, soluble Lasso-D clearly functioned as an attractant of axonal elongation and/or steering.

Since LPHN1 is present on axonal growth cones (*Figure 1*, *Figure 1—figure supplement 1*), binds soluble Lasso (*Figure 2*, *Figure 2—figure supplement 1*) and is the strongest interacting partner of Lasso (*Boucard et al., 2014*; *Silva et al., 2011*), we hypothesized that LPHN1 may be involved in the observed Lasso-mediated attraction of axons (*Figure 4—figure supplement 1A*). To investigate this, hippocampal cultures from LPHN1 KO or WT mice (genotyping shown in *Figure 4—figure supplement 1B*) were exposed to a gradient of Lasso-D added to the Axonal Compartment. The

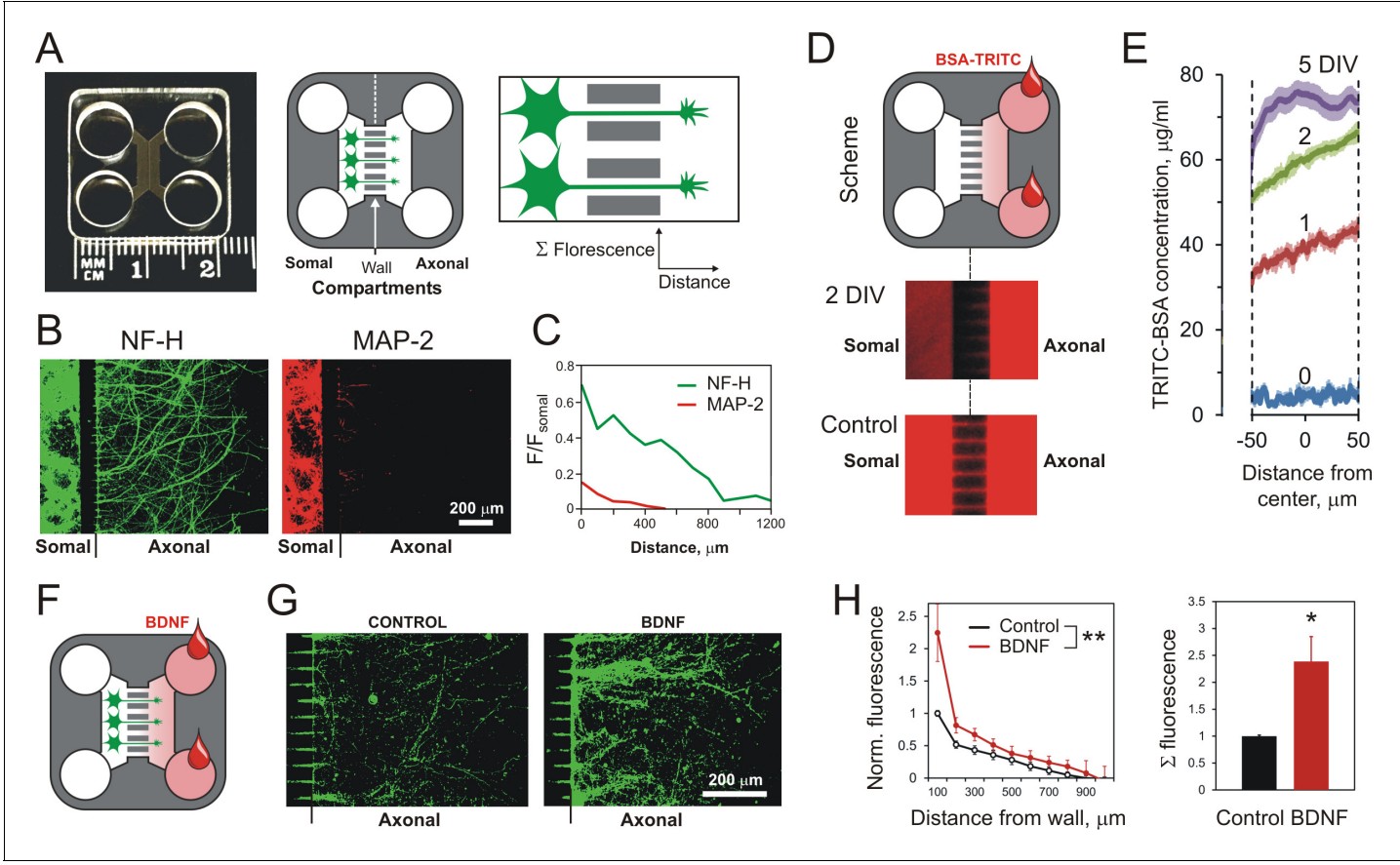

**Figure 3.** Using MAIDs to study axonal attraction by soluble chemoattractants. (**A**) Left, a photograph of a MAID. Center, a scheme of the experiment: neurons are seeded into the Somal Compartment and their neurites grow into the Axonal Compartment; both compartments are then stained for NF-H (axons) and MAP-2 (dendrites). Right, an enlarged portion of the separating wall showing the principles of fluorescence measurements in the Axonal Compartment. (**B**) Fluorescent images from the same MAID stained for NF-H (green) and MAP-2 (red) showing that axons penetrate into the Axonal Compartment significantly more readily than dendrites. (**C**) Profiles of NF-H and MAP-2 fluorescence in the Axonal Compartment, normalized to respective fluorescence in the Somal Compartment show that the relative degree of penetration of axons is ~5 fold higher compared to dendrites. (**D**) Gradients of soluble proteins can be established within microchannels and maintained for several days. Top, a scheme of the experiment: TRITC-conjugated BSA was added to the Axonal Compartment and monitored using time-lapse fluorescent microscopy. Middle, fluorescence distribution 2 days after TRITC-BSA addition. Bottom, fluorescence distribution after filling the whole MAID with TRITC-BSA. (**E**) Quantification of the TRITC-BSA gradient within microchannels (normalized to 100 μg/ml TRITC-BSA). The mean values are shown ±SEM; n = 4. (**F–H**) A gradient of BDNF in MAIDs acts as an axonal attractant. (**F**) A scheme of the experiment. (**G**) Representative images of NF-H-positive axons in the Axonal Compartment exposed to control conditions (left) or to a BDNF gradient in the microchannels (right). H. Left, Average profiles of normalized NF-H fluorescence in the presence or absence of BDNF (2-way ANOVA: **, p=0.002; $F_{1,84}$ = 10.15). Right, integrated NF-H fluorescence between 0 and 500 μm from the separating wall (t-test: *, p=0.04; n = 5).

DOI: https://doi.org/10.7554/eLife.37935.011

The following source data is available for figure 3:

**Source data 1.** Source data for *Figure 3*, Panels C, E, and H.
DOI: https://doi.org/10.7554/eLife.37935.012

total amounts of neurites and cells in both compartments were quantified using the lipophilic membrane tracer DiO (see Materials and methods for details). The results clearly demonstrated that the neurites from LPHN1-expressing (WT) hippocampal neurons crossed into the Lasso D-containing Axonal Compartment 5.5-fold more readily than the neurites from neurons lacking this receptor (*Figure 4E*, left). Importantly, this effect was not due to a lower viability of LPHN1 KO neurons, because there was no difference between the KO and WT cells within the Somal Compartment (*Figure 4E*, right).

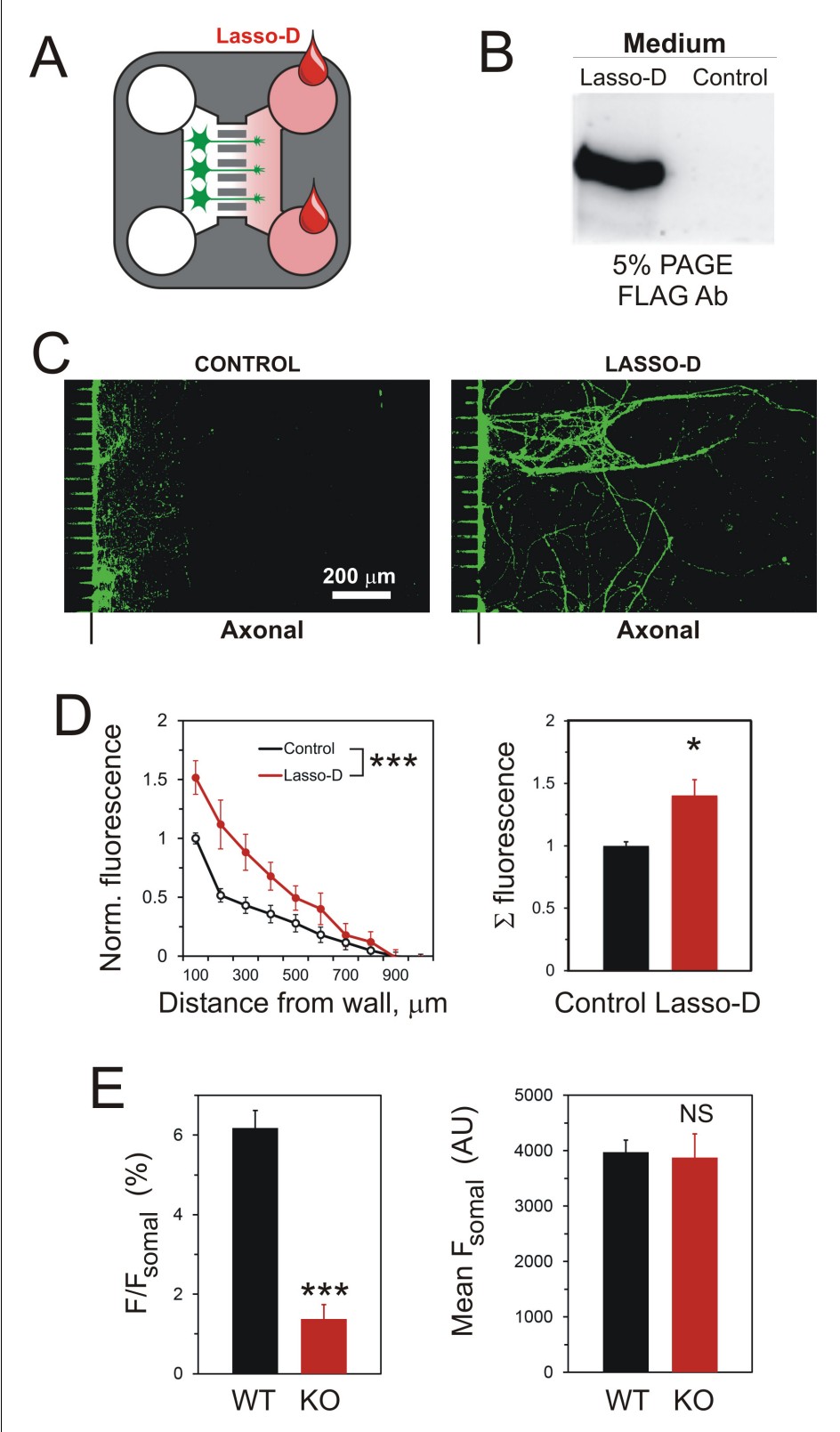

**Figure 4.** A gradient of soluble Lasso-D induces axonal attraction via LPHN1. (**A**) A scheme of the experiment: hippocampal neurons were cultured in Somal Compartments, purified Lasso was added to Axonal Compartments at 3 DIV. (**B**) Lasso remains intact in the Axonal Compartment. The media from Axonal Compartments were collected at 8 DIV and analyzed by Western blotting. (**C**). Images of NF-H-positive axons in the Axonal Compartment exposed to control medium (left) or Lasso-D (right). (**D**) Analysis of axonal growth in Axonal Compartments. Left, profiles of NF-H immunofluorescence
*Figure 4 continued on next page*

*Figure 4 continued*

with and without Lasso-D (3-way ANOVA: ***, p<0.001; $F_{1,144}$ = 12.92). Right, average integrated immunofluorescence at 0–500 µm from the wall, with and without Lasso-D (t-test: *, p=0.027; n = 7). (E) Knockout of LPHN1 blocks axonal attraction by soluble Lasso. Hippocampal neurons from *Adgrl1*[-/-] (LPHN1 KO) and *Adgrl1*[+/+] (LPHN1 WT) mice were cultured in MAIDs and exposed to Lasso-D gradient. The amount of cellular material in each compartment was quantified by DiO labeling at 8 DIV. E. Left, LPHN1 KO cultures sent significantly fewer neurites to Lasso-containing Axonal Compartments compared to WT cultures (t-test: ***, p<0.001, n = 3). **Right**, there was no difference in the number of cells, dendrites and axons in the Somal Compartments between the two types of cultures (t-test: N.S., p=0.4, n = 3).
DOI: https://doi.org/10.7554/eLife.37935.013

The following source data and figure supplement are available for figure 4:

**Source data 1.** Source data for *Figure 1—figure supplement 1*, Panels D and E.
DOI: https://doi.org/10.7554/eLife.37935.014

**Figure supplement 1.** Knockout of LPHN1 prevents axonal attraction by soluble Lasso.
DOI: https://doi.org/10.7554/eLife.37935.015

We also studied the behavior of axons in response to a spatio-temporal Lasso gradient in the corridor of the Axonal Compartment, by exposing axons to an increasing concentration of the attractant during the whole growth process. In order to achieve a stable increase in protein concentration over time, we seeded HEK293A cells stably expressing soluble Lasso-D (untransfected HEK293A cells were used in control) into the wells of the Axonal Compartment (*Figure 5A*). The presence of secreted Lasso-D within the Axonal Compartments was verified at the end of each experiment (*Figure 5B*), and the distribution of axons was quantified by NF-H immunofluorescence (*Figure 5C, D*). In this experiment, we observed not only a significantly greater number of axons being attracted, but also axons growing deeper into the corridors of the Axonal Compartments (*Figure 5D*). On the other hand, quantification of MAP-2 immunofluorescence demonstrated that released Lasso-D did not attract dendrites; in fact, there was a slight repulsive effect (*Figure 5E*). Taken together, these experiments indicate that a gradient of the soluble Lasso fragment specifically induces axonal attraction.

Soluble Lasso fragment also induced strong axonal fasciculation (e.g. *Figures 4C* and *5C*). This effect was quantified by measuring the width of axonal bundles at 100 µm from the separating wall, where axons grew mostly away from the wall rather than along it. Based on the average width of a single axon (1 µm), an average bundle contained 2–3 axons in control conditions, but more than five axons in the presence of 1.5 nM Lasso-D (*Figure 5F*). Thus, Lasso fragment can induce axonal fasciculation in a concentration-dependent manner.

In order to rule out the possibility that the observed effects of the released Lasso fragment were due to a general positive trophic effect (e.g. an increase in axonal elongation speed), Lasso-D was added directly to cultures of hippocampal neurons. To visualize axons, neurons were transfected with GFP prior to plating and allowed to grow for 4 DIV, after which the longest neurites of GFP-positive neurons were traced and measured. We did not detect any increase in the length of neurites when neurons were exposed to Lasso-D (*Figure 5G,H*).

Taken together, these data demonstrate unequivocally that a gradient of the soluble fragment of Lasso acts as an axonal attraction cue without affecting their overall growth.

## The mechanism of axonal attraction by lasso

To determine the downstream effects of the interaction between soluble Lasso ECD and LPHN1, we used neuroblastoma cells stably expressing LPHN1. It was reported previously that the signaling machinery downstream of LPHN1 in these cells is similar to that in neurons (*Silva et al., 2009*; *Volynski et al., 2004*). When the LPHN1-expressing neuroblastoma cells are stimulated by the known LPHN1 ligand and potent secretagogue LTX[N4C], the N-terminal and C-terminal fragments (NTF and CTF) of LPHN1 undergo rearrangement (as illustrated in *Figure 6A*, middle). In turn, this induces intracellular $Ca^{2+}$ signaling which involves the activation of $G\alpha_q$ and phospholipase C (PLC), and release of inositol 1,4,5-trisphosphate ($IP_3$) (*Silva et al., 2009*; *Volynski et al., 2004*).

These observations suggested that Lasso might also affect the distribution of NTF and CTF of LPHN1 in the plasma membrane. Indeed, we noticed that soluble Lasso-D or Lasso-A caused the NTF to aggregate into patches on the surface (*Figure 2C*, panel 2; *Figure 2—figure supplement 1C*). To test whether Lasso also causes a redistribution of the CTF required for intracellular signaling,

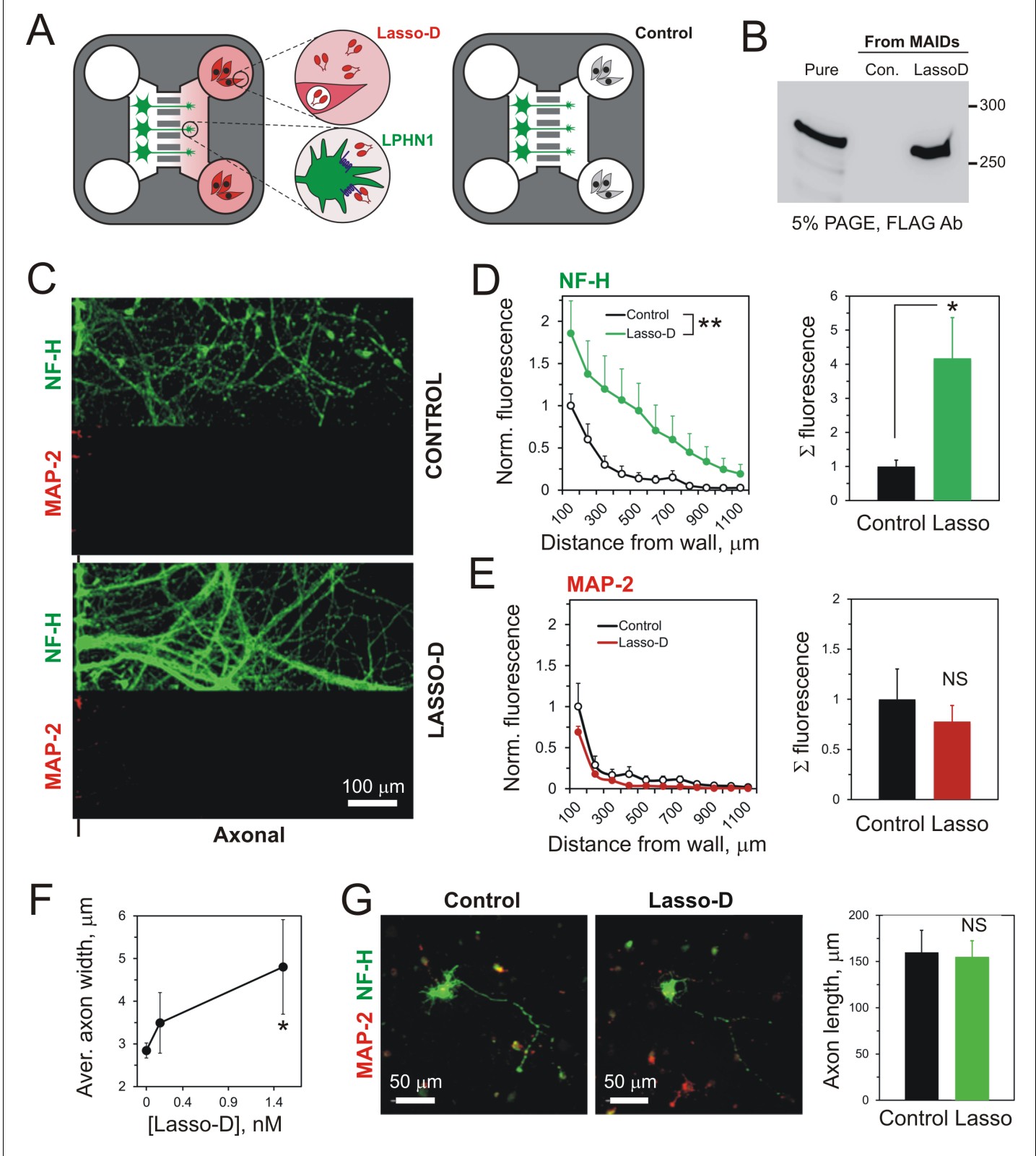

**Figure 5.** A spatio-temporal gradient of soluble Lasso induces axonal attraction and fasciculation, but does not increase axonal length. (**A**) A scheme of the experiment: HEK293A cells stably transfected with Lasso-D were cultured in the wells of Axonal Compartments; untransfected cells were used as a control. (**B**) A representative Western blot of the media from Axonal Compartments; Lasso-D is secreted by transfected HEK293A cells only and is stable. (**C**) Images of NF-H-positive axons (green) and MAP-2-positive dendrites (red) in the Axonal Compartment exposed to temporal gradients

*Figure 5 continued on next page*

Figure 5 continued

formed by control cells (top) or Lasso-D-expressing cells (bottom). (D) Left, profiles of axons in Axonal Compartments, identified by NF-H immunofluorescence, exposing a difference between control and Lasso-secreting cells (3-way ANOVA: **, p=0.006; n = 7, $F_{1,84}$ = 7.89). Right, average integrated axonal fluorescence at 0–500 μm from the wall, with control or Lasso-secreting cells (t-test: *, p=0.045; n = 7). (E) Left, profiles of dendrites in Axonal Compartments, identified by MAP-2 immunofluorescence, with control or Lasso-secreting cells (3-way ANOVA: non-significant, p=0.23; $F_{1,84}$ = 1.46). Right, average integrated dendritic fluorescence at 0–500 μm from the wall, with control or Lasso-secreting cells (t-test: non-significant, p=0.54; n = 7). (F) Soluble released Lasso-D induces axonal fasciculation. The width of all NF-H-positive axonal bundles was measured at 100 μm from the separating wall. The degree of fasciculation correlates with Lasso concentration (Pearson's correlation: $R^2$ = 0.43, p=0.041). (G) Soluble Lasso has no effect on axon length in cultured hippocampal cells. Left. Representative images of GFP-positive neurons immunostained for GAP-43 (red); after treatment with control medium (left) or with Lasso-D (right). Right. Quantification of the total neurite length in GFP-expressing neurons after the treatment (t-test: non-significant, p>0.05, n = 30 cells without Lasso-D and 61 cells with Lasso-D from three independent cultures).
DOI: https://doi.org/10.7554/eLife.37935.016
The following source data is available for figure 5:

Source data 1. Source data for *Figure 5*, Panels D-G.
DOI: https://doi.org/10.7554/eLife.37935.017

we applied Lasso-D to LPHN1-expressing cells and followed the fate of both NTF and CTF. We observed a dramatic rearrangement of both LPHN1 fragments in the membrane, leading to the formation of large molecular aggregates also containing Lasso (*Figure 6C*). Similar clustering of both LPHN1 fragments was also induced by LTX$^{N4C}$, a strong LPHN1 agonist (*Figure 6D*). On the other hand, an antibody recognizing the V5 epitope at the N-terminus of NTF only caused NTF clustering, but did not affect the distribution of CTF (*Figure 6A*, right; *Figure 6E*). Thus, soluble Lasso ECD, which causes the association of the LPHN1 fragments, might be a functional agonist of LPHN1, similar to LTX$^{N4C}$. By analogy, this also indicated that the soluble Lasso fragment could induce signal transduction via the CTF of LPHN1 coupled to a G-protein.

The effect of LTX$^{N4C}$ can be assessed by monitoring cytosolic Ca$^{2+}$ (*Silva et al., 2011*; *Volynski et al., 2004*). We therefore investigated whether the soluble Lasso ECD could induce similar effects. LPH1-expressing neuroblastoma cells were stimulated with saturating concentrations of Lasso-D, LTX$^{N4C}$ (positive control) or buffer (negative control), while cytosolic calcium levels were monitored using an intracellular Ca$^{2+}$-sensing dye, Fluo-4 (see *Figure 7—figure supplement 1A* for the scheme of experiment). Similar to LTX$^{N4C}$, in the absence of extracellular Ca$^{2+}$, Lasso-D did not cause any Ca$^{2+}$ signals in LPHN1-expressing NB2a cells (*Figure 7A*). However, when extracellular Ca$^{2+}$ was added to the cells, the rise in intracellular Ca$^{2+}$ signal was significantly higher in the presence of the ECD of Lasso, compared to negative control (*Figure 7A*). Thus, Lasso-D is able to cause intracellular Ca$^{2+}$ signaling in LPHN1-expressing cells.

One of the features of LTX$^{N4C}$-induced effects (such as Ca$^{2+}$ signaling and neurotransmitter release) is that they develop with a delay of ~20 min, which has been attributed to the time taken by the toxin to assemble the LPHN1 fragments together and cause its maximal activation (*Volynski et al., 2004*). We predicted, therefore, that the rearrangement of the NTF and CTF induced by soluble Lasso (*Figure 6C*) should prepare the signaling machinery for stimulation by the toxin. To test this idea, we first treated the LPHN1-expressing cells with Lasso-D and then with LTX$^{N4C}$ (*Figure 7—figure supplement 1B*). When Lasso-D was applied in the presence of 2 mM Ca$^{2+}$, it induced relatively short-lived intracellular Ca$^{2+}$ signaling (*Figure 7B*, right, prior to the blue arrowhead). However, when LTX$^{N4C}$ was then added, it triggered Ca$^{2+}$ signaling after a shorter delay (~14 min), instead of the usual ~23 min (*Figure 7C*). This additivity of effects is consistent with soluble Lasso inducing intracellular Ca$^{2+}$ signaling via the same molecular mechanism as LTX$^{N4C}$.

Another well-known effect of LTX$^{N4C}$ is the burst-like release of neurotransmitters, linked to the elevated levels of cytosolic Ca$^{2+}$ (*Lelyanova et al., 2009*; *Volynski et al., 2003*). As Lasso-D likewise increased intracellular Ca$^{2+}$ concentration, it might also trigger such transmitter exocytosis. To test this hypothesis, we applied a previously characterized (*Silva et al., 2011*), soluble, short C-terminal Lasso construct (Lasso-G, *Figure 1A*) to mouse neuromuscular preparations and recorded the spontaneous miniature end plate potentials (MEPPs), which correspond to individual exocytotic events. We found that incubation with Lasso-G significantly increased MEPPs frequency from 1.61 ± 0.27 Hz in control to 3.83 ± 0.79 Hz in the presence of Lasso-G (*Figure 7D,E*). However, this was much less than the effect of LTX$^{N4C}$, which triggered massive secretion of neurotransmitter reaching 29.5 ± 4.1

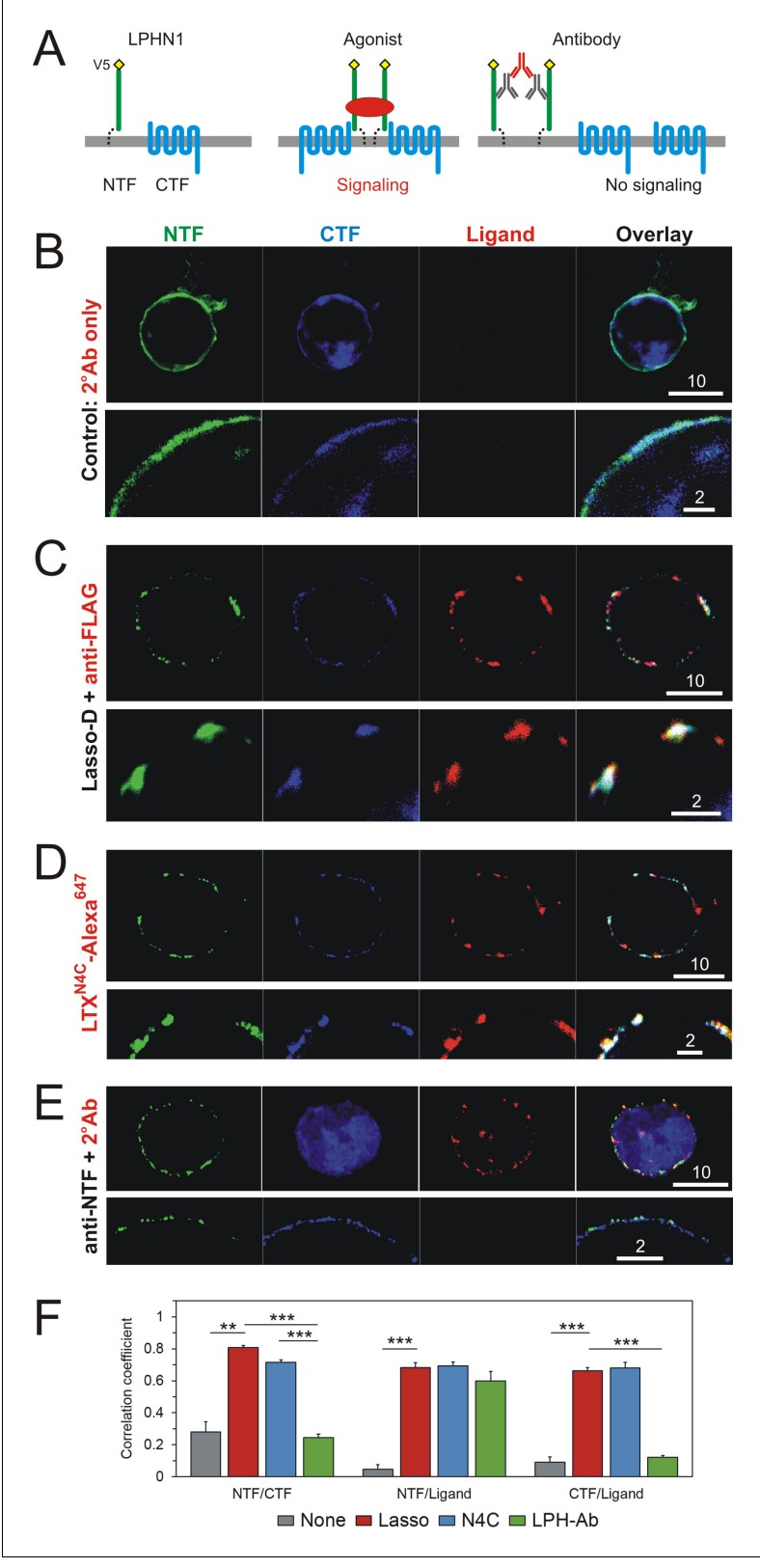

**Figure 6.** Interaction of LPHN1 with soluble Lasso causes LPHN1 aggregation. (**A**) A scheme of behavior of LPHN1 fragments at rest (left) and after binding an active agonist (middle) or a non-agonistic antibody (right). (**B–D**) Distribution of NTF and CTF in NB2a cells stably expressing LPHN1 and treated with control buffer (**B**), Lasso-D (**C**) or LTX[N4C] (**D**). (**E**) The binding of a non-agonistic antibody against NTF of LPHN1 does not cause an association of

*Figure 6 continued on next page*

*Figure 6 continued*

the NTF and CTF of LPHN1. Images shown are representative of 4 independent experiments (*n* = 4–7). All scale bars are in μm. (**F**) Quantitative analysis of correlation between the ligand-induced redistribution of NTF, CTF and ligand. T-test with Bonferroni correction: **, p<0.01; ***, p<0.001; *n* = 4–7 independent experiments.
DOI: https://doi.org/10.7554/eLife.37935.018
The following source data is available for figure 6:

**Source data 1.** Source data for *Figure 6*, Panel F.
DOI: https://doi.org/10.7554/eLife.37935.019

Hz (*Figure 7F*). To ascertain that both these effects were mediated by LPHN1, we used neuromuscular preparations from LPHN1 KO mice. Interestingly, unstimulated LPHN1 KO motor neurons showed an increased MEPPs frequency compared to synapses from WT animals (3.33 ± 0.79 Hz in KO synapses). However, neither Lasso-G, nor LTX$^{N4C}$ had any effect on exocytosis in preparations lacking LPHN1 (*Figure 7E,F*; 3.4 ± 0.68 Hz with Lasso-G and 3.8 ± 1.4 Hz with LTX$^{N4C}$). In all the recordings, the mean *amplitudes* of MEPPs under any condition did not differ significantly (*Figure 7—figure supplement 1C*), which indicated a purely presynaptic effect of the two LPHN1 agonists and of LPHN1 ablation. These results show that the soluble Lasso fragment can increase exocytosis at nerve terminals, and confirm the importance of LPHN1 in the observed effects of LTX and the ECD of Lasso.

From the results reported here, we hypothesize that the soluble Lasso fragment, released by developing neurons, interacts with LPHN1 on axonal growth cones and nerve terminals. It then induces clustering of LPHN1 fragments and activation of downstream signaling, causing an increase in cytosolic Ca$^{2+}$ and subsequent exocytosis. The latter two processes are known to be key regulators of axonal attraction (*Tojima et al., 2011*). Thus, the ability of soluble Lasso to activate these processes on axonal growth cones could underpin the mechanisms by which it attracts axons.

## Discussion

This study provides evidence that Lasso (a splice variant of TEN2 lacking a 7-residue insert in the β-propeller domain, TEN2-SS) functions specifically as an attractant for axons expressing LPHN1, and proposes a molecular mechanism for this effect. By using microfluidic devices to create long-term gradients of soluble proteins (*Figure 3*), we demonstrate that a gradient of soluble ECD of Lasso can act as an attractant for axons from hippocampal neurons (*Figures 4* and *5A–E*). Importantly, growing hippocampal neurons in a medium containing a uniform concentration of Lasso had no effect on the length of their axons (*Figure 5G*). This shows that Lasso plays an instructive role in the directionality, rather than the amount, of axonal growth. This is consistent with the effect of other axon attractants acting via similar mechanisms. For example, short-term exposure of axonal growth cones to gradients of BDNF stimulates IP$_3$-induced Ca$^{2+}$ release (IICR) that causes axonal attraction without an overall effect on neurite extension (*Li et al., 2005*).

One interesting observation from this project was the fasciculation of neurites in response to soluble Lasso/TEN2 (*Figure 5C,F*). Fasciculation of axons is one of the major mechanisms of axonal navigation, for example in limb development (*Bastiani et al., 1986*). While axonal fasciculation has not been previously linked to a soluble ECD of TEN, neurite bundling was actually observed in hippocampal cultures in response to TEN1 C-terminal peptide (TCAP-1) (*Al Chawaf et al., 2007*). Furthermore, knockdown of TEN1 in *C. elegans* resulted in de-fasciculation of the axons in the ventral nerve cord (*Drabikowski et al., 2005*). Potential mechanisms of axonal bundling include actin reorganization induced by an LPHN1-mediated rise in cytosolic Ca$^{2+}$, other unknown interactions with cell adhesion molecules, or it could also be due to the divalent Lasso/TEN2 fragment crosslinking adjacent axons, thus promoting their parallel elongation.

The soluble Lasso/TEN2 fragment could potentially have two membrane-anchored receptors: (i) TEN2 itself, as a homophilic ligand (*Bagutti et al., 2003*; *Rubin et al., 2002*), or (ii) LPHN1, as a heterophilic ligand (*Boucard et al., 2014*; *Silva et al., 2011*). However, we have not observed TEN2 expression in growth cones of hippocampal axons (*Figure 1E*), but found it to be abundant on

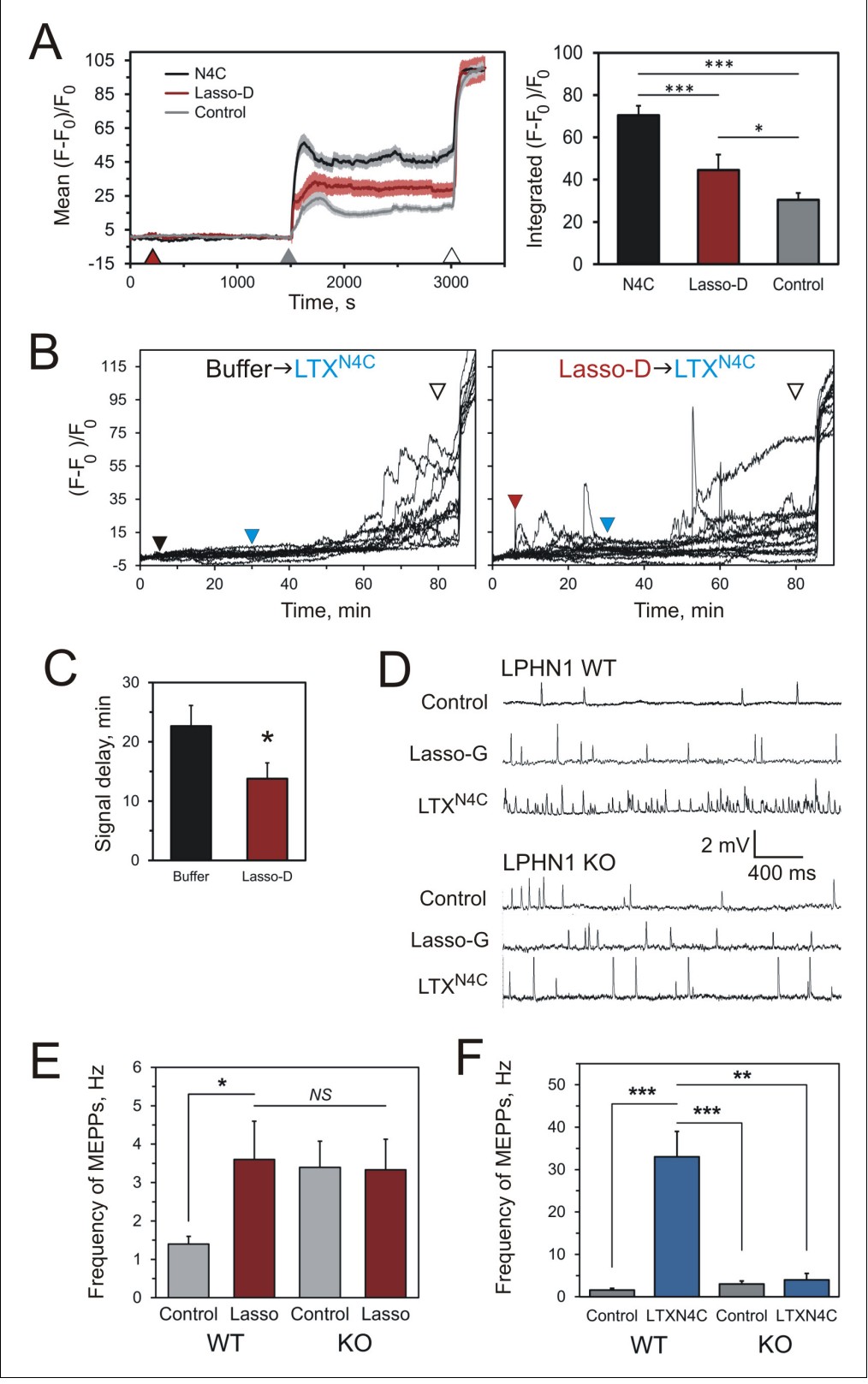

**Figure 7.** Soluble Lasso induces Ca2+signaling in LPHN1-expressing cells and enhances spontaneous exocytosis at neuromuscular junctions. (**A**) Changes in intracellular $Ca^{2+}$ concentration in neuroblastoma cells stably expressing LPHN1 were monitored using a $Ca^{2+}$ indicator dye, Fluo-4. The scheme of the experiment is shown in *Figure 7—figure supplement 1A*. After 5 min recording of baseline fluorescence, the cells were treated (maroon
*Figure 7 continued on next page*

*Figure 7 continued*

arrowhead) with control buffer, 1 nM LTX$^{N4C}$ or 360 nM Lasso-D. 20 min later, 2 mM Ca$^{2+}$ was added (gray arrowhead) to synchronize the intracellular Ca$^{2+}$ signaling, followed by 1 nM wild-type α-latrotoxin (open arrowhead) to measure F$_{max}$, for normalization. Left, profiles of normalized Fluo-4-Ca$^{2+}$ fluorescence over time for the three conditions used (mean values ± SEM are shown; the data are from 80 to 120 individual cells from *n* = 4 independent experiments). Right, integration of Fluo-4-Ca$^{2+}$ fluorescence over time (from *B*). Pre-treatment with Lasso-D potentiates intracellular Ca$^{2+}$ signaling. T-test with Bonferroni correction: *, p<0.05; ***, p<0.001. (**B**) Experiments testing the effect of Lasso-D on the time-course of LTX$^{N4C}$-induced LPHN1-dependent Ca$^{2+}$ signaling. Cells expressing LPHN1 were loaded with Fluo-4 and stimulated first with control buffer (black arrowhead, left) or 1.5 nM Lasso-D (maroon arrowhead, right), and then with 2 nM LTX$^{N4C}$ (blue arrowhead). 1 nM wild-type LTX was added at the end (open arrowhead). Ca$^{2+}$ fluorescence measurements were obtained as in *A*. Representative normalized Ca$^{2+}$ fluorescence profiles are shown. (**C**) Time delay before the onset of LTX$^{N4C}$-induced signaling in cells pretreated with control buffer or Lasso-D determined from traces in B. T-test: *, p<0.05; the data are from 166 buffer-LTX$^{N4C}$-treated cells and from 144 Lasso-LTX$^{N4C}$-treated cells, from *n* = 5 independent experiments. (**D**) Representative raw recordings of MEPPs in neuromuscular preparations from LPHN1 WT and KO mice, in buffer containing 2 mM Ca$^{2+}$ without any agonists or in the presence of 20 nM Lasso-G or 1 nM LTX$^{N4C}$. (**E**) The frequency of MEPPs in the absence or presence of 20 nM Lasso-G, as in D. Lasso-G significantly increases the frequency of MEPPs at neuromuscular junctions from WT mice, but has no effect on exocytosis in LPHN1 KO synapses. The data shown are the means ± SEM from 21 (control) and 23 (Lasso-G) individual muscle fibers from 5 WT preparations and 36 and 26 muscle fibers from 6 KO preparations. (F) Positive control: 1 nM LTX$^{N4C}$ increases the frequency of MEPPs in WT, but not in LPHN1 KO neuromuscular junctions. The data are the means ± SEM from 21 and 32 individual muscle fibers from 6 WT preparations and 36 and 12 muscle fibers from 6 KO preparations. Mann-Whitney test with Bonferroni correction for multiple comparisons: *, p<0.05; **, p<0.01; ***, p<0.001; NS, non-significant.
DOI: https://doi.org/10.7554/eLife.37935.020

The following source data and figure supplements are available for figure 7:

**Source data 1.** Source data for *Figure 7*, Panels A-C, E, and F.
DOI: https://doi.org/10.7554/eLife.37935.021

**Figure supplement 1.** Design of the experiments testing Lasso induced Ca2+signaling in LPHN1-expressing cells and its presynaptic action at mouse neuromuscular junctions.
DOI: https://doi.org/10.7554/eLife.37935.022

**Figure supplement 1—source data 1.** Source data for *Figure 7—figure supplement 1*, Panel C.
DOI: https://doi.org/10.7554/eLife.37935.023

---

dendrites (*Silva et al., 2011*) (*Figure 1E*, *Figure 1—figure supplement 1B*). We also did not detect any appreciable binding of the released Lasso ECD to membrane-anchored Lasso (*Figure 2D*, *Figure 2—figure supplement 1B*). In addition, homophilic interaction of Lasso/TEN2 actually has been reported to inhibit neurite outgrowth in neuroblastoma cells (*Beckmann et al., 2013*), while we saw an opposite effect (*Figures 4* and *5*). Thus, the potential Lasso/TEN2 homophilic interaction could not explain the observed axonal attraction. On the other hand, we found strong expression of LPHN1 on the axonal growth cones of cultured hippocampal neurons (*Figure 1E–I*, *Figure 1—figure supplement 1C–F*) (*Silva et al., 2011*). Importantly, the released soluble ECD of Lasso strongly bound to LPHN1 that was expressed on neuroblastoma cells or neuronal growth cones (*Figure 2*, *Figure 2—figure supplements 1–2*). Furthermore, we found that deletion of LPHN1 precluded axonal attraction by Lasso (*Figure 4*), while it had no effect on neuronal cell bodies and dendrites in the Somal Compartment. These data strongly implicate LPHN1 in mediating Lasso-induced axon attraction.

Our studies also reveal the likely mechanism that underlies the Lasso/LPHN1-induced axonal attraction. LPHN1 is a G-protein-coupled receptor (GPCR) that physically and functionally links to Gα$_{q/11}$ (*Rahman et al., 1999*). Activation of LPHN1 by its non-pore-forming agonist, LTX$^{N4C}$, leads to aggregation of the NTF and CTF of LPHN1 (*Silva et al., 2009*; *Volynski et al., 2004*). This results in assembly of a functional GPCR, with subsequent activation of the downstream signaling cascade, which includes Gα$_{q/11}$, phospholipase C, production of IP$_3$ and IP$_3$-receptor-mediated release of Ca$^{2+}$ from intracellular stores (*Capogna et al., 2003*; *Lajus et al., 2006*; *Volynski et al., 2004*), thus inducing IICR.

IICR is also regulated and enhanced by increased cAMP levels (*Tojima et al., 2011*), and we previously demonstrated that activation of LPHN1 expressed in COS7 cells induces an increase in cAMP production (*Lelianova et al., 1997*). In line with this, the recent study by *Li et al. (2018)* confirmed the ability of LPHN1 to regulate cAMP signaling. In that work (*Li et al., 2018*), the cAMP signaling interference system was based on HEK293 cells expressing exogenous $\beta_2$ adrenoceptor ($\beta$2AR). Activation of $\beta$2AR by its agonist led to an increase in cAMP production, while a large excess of co-expressed LPHN1 interfered with $\beta$2AR signaling. This clearly suggests that LPHN1 uses the same cAMP signaling machinery as $\beta$2AR, and that when LPHN1 is not stimulated, it can titrate components of this machinery, decreasing their availability to $\beta$2AR.

In agreement with the role of Lasso as a functional LPHN1 agonist, the binding of the released Lasso fragment to LPHN1 similarly causes the re-association of LPHN1 fragments (*Figure 6*) and $Ca^{2+}$ signaling (*Figure 7A–C*). A rise in cytosolic $Ca^{2+}$ concentration, in turn, can increase the rate of exocytosis, and we indeed observed enhanced acetylcholine release in mouse neuromuscular junctions in response to soluble Lasso (*Figure 7D–F*). This response to Lasso was clearly mediated by LPHN1, as it was not detected in neuromuscular preparations from LPHN1 KO mice (*Figure 7D–F*). On the other hand, the effect of soluble Lasso on vesicular exocytosis was much weaker – and probably more physiological – than the massive effect of $LTX^{N4C}$.

In addition to $Ca^{2+}$ regulation, Lasso binding to LPHN1 can induce cAMP signaling. Indirect evidence for this is provided by the cAMP signaling interference experiments mentioned above (*Li et al., 2018*). When LPHN1 co-expressed with $\beta$2AR was stimulated for 24 hr with Lasso/TEN2 (expressed on the same or opposite cells), this strongly decreased cAMP levels induced by $\beta$2AR activation. The most likely reason could be that, following an initial Lasso-induced LPHN1 activation, which normally subsides within 30 min (*Figure 7B*), the continued LPHN1 stimulation led to massive heterologous receptor desensitization (*Kelly et al., 2008*) and inhibition of $\beta$2AR-mediated effect.

Intriguingly, the effects of soluble Lasso resemble the well-known mechanism that underpins axonal attraction and consists of $IP_3$ receptor-mediated local release of $Ca^{2+}$ from intracellular stores, coupled with an increase in cAMP levels, that leads to increased exocytosis at the advancing edge of a growth cone (*Akiyama et al., 2009*; *Qu et al., 2002*; *Tojima et al., 2011*; *Tojima and Kamiguchi, 2015*). Thus, when a gradient of soluble Lasso ECD approaches one side of an axonal growth cone, it may cause local activation of LPHN1 and its downstream signaling, ultimately leading to IICR. Local IICR in growth cones can induce an increase in vesicular exocytosis (as observed in our experiments with Lasso-G, *Figure 7*) and the remodeling of actin filaments (*Tojima et al., 2011*). The resulting augmented membrane delivery and actin-driven extension of filopodia at the edge facing a Lasso gradient would support the growth cone's advance in this direction. Thus, based on all our data, we propose this chain of events (summarized in *Figure 8*) as a likely mechanism for axonal attraction by soluble Lasso observed in this study.

While TEN2 has been implicated in axon guidance in the visual pathway (*Young et al., 2013*), here we report that it can also trigger axonal steering in developing hippocampal neurons, which is consistent with the strong expression of both Lasso/TEN2 and LPHN1 in the hippocampus (*Davletov et al., 1998*; *Otaki and Firestein, 1999*). Furthermore, both proteins are expressed throughout the CNS, suggesting that this mechanism of soluble Lasso/LPHN1-mediated axonal attraction may apply widely across the brain, especially in such areas as the cortex, cerebellum, thalamus and spinal cord.

Interestingly, the splice variant of TEN2 (TEN2+SS), which contains the 7-amino acid insert in the $\beta$-propeller domain and cannot mediate cell adhesion via LPHN1 (*Li et al., 2018*), might attract dendrites instead of axons, in contrast to Lasso (TEN2-SS). Thus, in an artificial synapse formation experiment (*Li et al., 2018*), HEK293 cells expressing TEN2+SS were seen covered by neurites from co-cultured hippocampal neurons that contained $GABA_A$ receptors. However, these processes did not show a proportionate accumulation of PSD-95 and thus probably represented *en passant* dendrites that were attracted to TEN2+SS cells, but unable to form mature inhibitory synapses with them. This could be a mechanism by which TEN2+SS could provide a substrate for the growth of dendrites searching for their ultimate target/s. Although the relative abundance of Lasso and TEN2+SS in the brain is unknown, these data suggest that various TEN isoforms could participate in distinct interactions, possibly with opposite results.

High expression of LPHN1 and Lasso/TEN2 throughout the CNS, combined with their fundamental role in axon guidance, is consistent with lethal phenotypes observed in simpler organisms

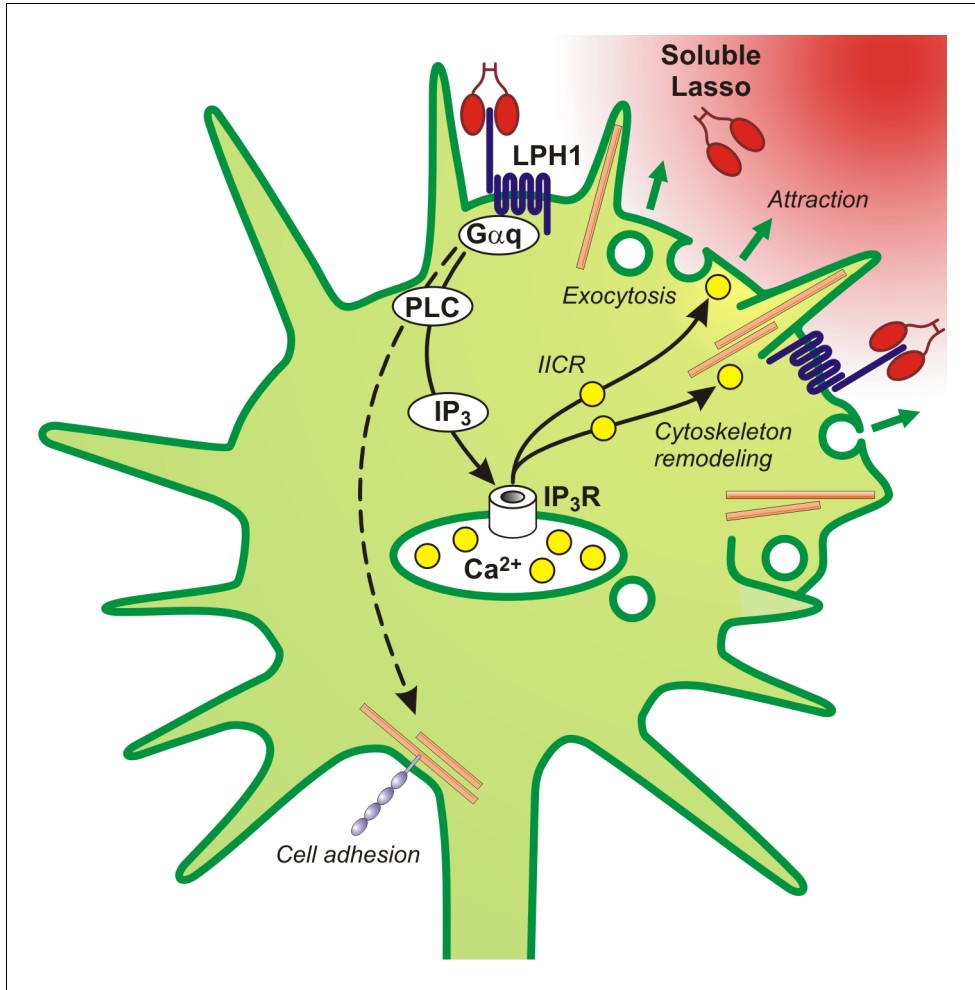

**Figure 8.** A proposed scheme of the mechanism of axonal attraction by released Lasso ECD. When Lasso binds the NTF of LPHN1, it causes its re-association with the CTF. This activates $G\alpha_{q/11}$ and triggers the PLC signaling cascade. Downstream of this cascade, the local $IP_3$-induced calcium release (IICR) from intracellular stores stimulates exocytosis and may also stimulate reorganization of actin through $Ca^{2+}$/calmodulin-dependent protein kinase II (CaMKII), thus mediating axonal attraction. The dashed line represents LPHN1-mediated activation of neuronal adhesion molecules via an unknown mechanism that may lead to axonal fasciculation observed in the presence of soluble Lasso (*Figure 5C, F*).

DOI: https://doi.org/10.7554/eLife.37935.024

(*Langenhan et al., 2009*; *Mosca et al., 2012*). In knockout mice, however, the phenotype is less severe (*Tobaben et al., 2002*; *Young et al., 2013*) (Ushkaryov, to be published elsewhere) suggesting that LPHN1 deletion is not completely penetrant, likely due to a compensatory effect of multiple LPHN and TEN homologs expressed in the mammalian brain. Indeed, LPHN1 can also weakly interact with TEN4 (*Boucard et al., 2014*), and LPHN3 can interact with TEN1 (*O'Sullivan et al., 2014*). Moreover, LPHN and TEN isoform expression patterns overlap (*Oohashi et al., 1999*; *Sugita et al., 1998*; *Zhou et al., 2003*). This predisposition to compensation further raises the possibility that the mechanism of axonal guidance involving the interaction of soluble TEN2 with LPHN1, described in this study, may occur between different members of the LPHN and TEN families. These observations provide evidence of further diversity of interactions and local specificity of developmental pathways for more accurate and plastic patterning of neural networks within the mammalian CNS.

# Materials and methods

## Key resources table

| Reagent type (species) or resource | Designation | Source or reference | Identifiers | Additional information |
|---|---|---|---|---|
| Antibody | Anti-FLAG M2 affinity gel | Sigma-Aldrich | A2220 | |
| Antibody | Chicken anti-myc | Millipore | AB3252 RRID:AB_2235702 | (Immunocytochemistry 1:1,000) |
| Antibody | Mouse anti-actinin | Sigma-Aldrich | A7811 | (Western blot 1:1,500) |
| Antibody | Mouse anti-FLAG M2 | Sigma-Aldrich | F3165 RRID:AB_259529 | (Immunocytochemistry 1:1,000) |
| Antibody | Mouse anti-Lasso/ teneurin-2 C-terminus | (*Silva et al., 2011*) | dmAb | TN2C (Immunocytochemistry 1:300; Western blot 1:1,000) |
| Antibody | Mouse anti-MAP-2 | Neuromics | MO22116 | (Immunocytochemistry 1:1,000) |
| Antibody | Mouse anti-synapsin | Santa-Cruz Biotechnology | sc-376623 RRID:AB_11150313 | (Immunocytochemistry 1:1,000) |
| Antibody | Mouse monoclonal anti-myc | Millipore | 05–419 RRID:AB_309725 | clone 9E10 (Immunocytochemistry 1:1000; Western blot 1:) |
| Antibody | Mouse monoclonal anti-V5 | AbD Serotec/Bio-Rad | MCA1360 | clone SV5-Pk1 (Immunocytochemistry 1:2,000) |
| Antibody | Rabbit anti-GFP | Thermo Fisher Scientific | A-11122 RRID: AB_221569 | (Immunocytochemistry 1:1,000) |
| Antibody | Rabbit anti-NF-H | Neuromics | RA22116 | (Immunocytochemistry 1:1,000; Western blot 1:10,000) |
| Antibody | Rabbit anti-PSD-95 | Millipore | AB9708 RRID:AB_11212529 | (Immunocytochemistry 1:2,000) |
| Antibody | Rabbit anti-Tau | Synaptic Systems | 314 002 RRID:AB_993042 | (Immunocytochemistry 1:1,000) |
| Antibody | Rabbit anti-V5 | Thermo Fisher Scientific | PA1-29324 RRID:AB_1961277 | (Immunocytochemistry 1:2,000) |
| Antibody | Rabbit polyclonal anti-LPHN1 NTF | (*Davletov et al., 1998*) | RL1 | (Immunocytochemistry 1:1,000) |
| Antibody | Rabbit polyclonal anti-LPHN1-peptide | (*Davydov et al., 2009*) | PAL1 | (Immunocytochemistry; Western blot 3 ng/mL) |
| Antibody | Sheep anti-teneurin-2 N-terminus | R and D systems | AF4578 RRID:AB_10719438 | TN2N (Western blot 1 µg/mL) |
| Cell line (*Homo sapiens*) | HEK293A | ECCC | RRID:CVCL_6910 | |
| Cell line (*Mus musculus*) | Neuroblastoma 2a | ATCC | RRID:CVCL_0470 | |
| Chemical compound | B27 Supplement | Life Technologies | 17504044 | |
| Chemical compound | Ca-free Hibernate-A medium | BrainBits UK | HE-Ca | |
| Chemical compound | Fluo-4 acetomethoxy ester | Thermo Fisher Scientific | F14201 | |
| Chemical compound | Insulin Transferrin Selenium Supplement | Life Technologies | 41400045 | |
| Chemical compound | Neurobasal-A medium | Thermo Fisher Scientific | 21103049 | |

*Continued on next page*

*Continued*

| Reagent type (species) or resource | Designation | Source or reference | Identifiers | Additional information |
|---|---|---|---|---|
| Chemical compound | Purified protein: BSA-TRITC | Thermo Fisher Scientific | A23016 | |
| Chemical compound | Vybrant DiO | Thermo Fisher Scientific | V22886 | |
| Commercial assay or kit | Amaxa Rat Neuron Nucleofector Kit | Lonza | VAPG-1003 | |
| Commercial assay or kit | SuperSignal West Femto Maximum Sensitivity Substrate | Thermo Fisher Scientific | 34094 | |
| Other | Microfluidic Axon Isolation Devices (MAIDs) | Xona Microfluidics | SND150 | |
| Recombinant DNA reagent | BLOCK-iT Lentiviral Pol II miR RNAi Expression System pLenti6/V5-GW/EmGFP-miR | Life Technologies | K4938-00 | |
| Recombinant DNA reagent | Bottom pre-miRNA oligo targeting LPHN1 mRNA | This paper | LPHN1miR14B | Sequence provided under Methods |
| Recombinant DNA reagent | Lasso-A | (*Silva et al., 2011*) | GenBank: JF784341 | |
| Recombinant DNA reagent | Lasso-D | (*Silva et al., 2011*) | GenBank: JF784344 | |
| Recombinant DNA reagent | Lasso-FS | (*Silva et al., 2011*) | GenBank: JF784340 | |
| Recombinant DNA reagent | Lasso-G | (*Silva et al., 2011*) | GenBank: JF784347 | GST-Lasso |
| Recombinant DNA reagent | LPH-42 | (*Volynski et al., 2004*) | GenBank:MF966512 | V5-LPH-A |
| Recombinant DNA reagent | pLenti6.2-GW/EmGFP-miR negative control | Thermo Fisher Scientific | K4938-00 | |
| Recombinant DNA reagent | Primer: N255: Neo Forward | This paper | | Sequence provided under Methods |
| Recombinant DNA reagent | Primer: N424: Neo/LPHN1 Reverse | This paper | | Sequence provided under Methods |
| Recombinant DNA reagent | Primer: N425: LPHN1 Forward | This paper | | Sequence provided under Methods |
| Recombinant DNA reagent | Top pre-miRNA oligo targeting LPHN1 mRNA | This paper | LPHN1miR14T | Sequence provided under Methods |
| Peptide, recombinant protein | Purified protein: Alexa Fluor 647-labeled LTX$^{N4C}$ | (*Volynski et al., 2004*) | N/A | |
| Peptide, recombinant protein | Purified protein: Human BDNF | R and D Systems | 248-BD | |
| Peptide, recombinant protein | Purified protein: Lasso-D | (*Silva et al., 2011*) | N/A | |
| Peptide, recombinant protein | Purified protein: Lasso-G | (*Silva et al., 2011*) | N/A | GST-Lasso |
| Peptide, recombinant protein | Purified protein: LTX$^{N4C}$ | (*Volynski et al., 2003*) | N/A | |
| Software | AxoScope 10 | Axon Instruments | | |

*Continued on next page*

*Continued*

| Reagent type (species) or resource | Designation | Source or reference | Identifiers | Additional information |
|---|---|---|---|---|
| Software | FIJI, ImageJ | NIMH, Bethesda, Maryland, USA | RRID:SCR_002285 RRID:SCR_003070 | |
| Software | LSM 510 Software (for image acquisition) | Carl Zeiss Microimaging GmbH | LSM 510 | |
| Software | LSM Image Browser (for image archiving and measurements) | Carl Zeiss Microimaging GmbH | RRID:SCR_014344 | |
| Software | MATLAB | Mathworks | RRID:SCR_001622 | |
| Software, algorithm | MATLAB | Mathworks | https://github.com/artificialbrain-tech/Axon-Guidance-Scripts | Axonal guidance scripts |
| Software | MiniAnalysis | Synaptosoft | | |
| Software | Volocity (for image acquisition and stitching) | Perkin-Elmer | RRID:SCR_002668 | |
| Strain (*Escherichia coli*) | *E. coli*: K12 JM109 | Promega Corporation | L2005 | |
| Strain (*Mus musculus*) | Mouse: C57BL/6J, *Adgrl1*$^{-/-}$, LPHN1 KO | This paper | AG148/2 | P0 hippocampus |
| Strain (*Mus musculus*) | Mouse: C57BL/6J, *Adgrl1*$^{-/-}$, LPHN1 KO | This paper | AG148/2 | P21 *flexor digitorum brevis* muscle |
| Strain (*Rattus norvegicus*) | Rat: E18 hippocampus | BrainBits UK | Rhp | |

## Chemical reagents

All chemicals and reagents were purchased from Sigma-Aldrich, unless otherwise stated. Cell culture reagents were from PAA Laboratories or Thermo Fisher Scientific. Purified proteins: LTX$^{N4C}$ (*Volynski et al., 2003*); LTX$^{N4C}$ labeled with Alexa Fluor 647 (*Volynski et al., 2004*); Lasso-G (*Silva et al., 2011*); Lasso-D (*Silva et al., 2011*) were prepared in this laboratory; human BDNF was from R&D Systems (248-BD); BSA-TRITC, from Thermo Fisher Scientific (A23016).

## Antibodies

The following antibodies were used in this work: Rabbit anti-NF-H (Neuromics, RA22116); mouse anti-MAP-2 (Neuromics, MO22116); mouse monoclonal anti-V5 (clone SV5-Pk1, AbD Serotec/Bio-Rad, MCA1360); rabbit anti-V5 (Thermo Fisher Scientific, PA1-29324; RRID:AB_1961277); mouse monoclonal anti-myc (clone 9E10, Millipore, 05–419; RRID:AB_309725); chicken anti-myc (Millipore, AB3252; RRID:AB_2235702); mouse anti-FLAG M2 (Sigma-Aldrich, F3165; RRID:AB_259529); anti-FLAG M2 affinity gel (Sigma-Aldrich, A2220); mouse anti-actinin (Sigma-Aldrich, A7811); rabbit poly-clonal anti-LPHN1-peptide (PAL1, (*Davydov et al., 2009*); rabbit polyclonal anti-LPHN1 NTF (RL1) (*Davletov et al., 1998*); mouse anti-Lasso/TEN2 C-terminus (TN2C, dmAb) (*Silva et al., 2011*); sheep anti-TEN2 N-terminus (TN2N, R and D systems, AF4578; RRID:AB_10719438); mouse anti-syn-apsin (Santa-Cruz Biotechnology, sc-376623; RRID:AB_11150313); rabbit anti-PSD-95 (Millipore, AB9708; RRID:AB_11212529); rabbit anti-Tau (Synaptic Systems, 314 002; RRID:AB_993042); rabbit anti-GFP (Thermo Fisher Scientific, A-11122; RRID: AB_221569).

## Cell lines

The following cell lines were used: human embryonic kidney cells (HEK293A, purchased from ECCC; RRID:CVCL_6910); mouse neuroblastoma cells (NB2a, a kind gift from Dr. C. Isaac, Imperial College London; originally from ATCC and subsequently authenticated by ATCC using their proprietary methods.; RRID:CVCL_0470). Both cultures are mycoplasma-free, based on a mycoplasma test kit PlasmoTest (Invivogen).

## Animals and biological samples

A LPHN1 KO mouse (strain AG148-2, *Adgrl1*[-/-]) was generated on the 129SvJ genetic background. Briefly (details to be published elsewhere), the LPHN1 gene was isolated from a BAC clone containing a 36-kbp fragment of mouse genomic DNA. This was used to design a transfer vector for homologous recombination, containing a 13-kbp gene fragment of the LPHN1 gene, in which the intron between exons 1 and 2 was replaced with a neomycin gene/promoter cassette flanked by two loxP sequences. This insert disrupted the open reading frame in the mRNA transcribed from the resulting mutated LPHN1 gene. The transfer vector, carrying also a negative selection marker (diphtheria toxin A-chain), was used to generate stably transfected 129Sv/J ES cell lines and chimeric mice, using standard transgenic techniques. Mice transmitting the inactivated LPHN1 gene through the germline were selected, inbred, back-crossed onto C57BL/6J background, and maintained at Charles River UK. LPHN1 gene disruption was confirmed by Southern blotting, PCR amplification using multiple primer pairs and Western blotting. The genotype of all animals used for breeding and tissue extraction was determined by PCR. All procedures (breeding and Schedule 1) were approved by the University of Kent Animal Welfare Committee and performed in accordance with Home Office regulations and the European Convention for the Protection of Vertebrate Animals used for Experimental and Other Scientific Purposes.

E18 hippocampi were obtained from rats (BrainBits UK, Rhp). P0 hippocampi were prepared from P0 mice (strains: C57BL/6J, *Adgrl1*[+/+], LPHN1 WT, or AG148/2, *Adgrl1*[-/-], LPHN1 KO). *Flexor digitorum brevis* muscle preparations were isolated from P21 male mice (C57BL/6J or AG148/2).

## Molecular biology reagents

The sequences of human Lasso (Ten–2) mutants used in this study are available at GenBank: Lasso-FS (JF784340), Lasso-A (JF784341), Lasso-D (JF784344), GST-Lasso (JF784347). N- and C-terminally tagged rat LPHN1 (termed also LPH-42, MF966512) was described previously as V5-LPH-A (*Volynski et al., 2004*). All cDNAs were subcloned into the pcDNA3.1 vector (Thermo Fisher Scientific). A negative control plasmid, pLenti6.2-GW/EmGFP-miR (Thermo Fisher Scientific, K4938-00), was used for GFP expression, and the miRNA oligonucleotides listed below were cloned into this vector for LPHN1 knock-down experiments.

Oligonucleotides for targeting LPHN1 mRNA were: LPHN1miR14T, (TGCTGATAAAC AGAGCG-CAGCACATAGTTTTGGCCACTGACTGACTATGTGCTGCTCTGTTTAT) and LPHN1miR14B (CCTGA TAAACAGAGCAGCACATAGTCAGTCAGTGGCCAAAACTATGTGCT GCGCTCTGTTTATC). PCR primers for genotype analysis were: Neo Forward (N255, CGAGACTAGTGAGACGTGCTACTTCCA TTTGTC); LPHN1 Forward (N425, CTGACCCATA ACCTCCAAGATGATGTTTAC); Neo/LPHN1 Reverse (N424, GATCTTGTCA TCTGTGCGCCCGTA).

## Generation of stable cell lines

Human embryonic kidney (HEK293A) and rat neuroblastoma (NB2a) cell lines were cultured using standard techniques in DMEM with 10% heat-inactivated fetal bovine serum (FBS, PAA Laboratories), at 5% $CO_2$ and 37°C. Stable cell lines were generated using the Escort III transfection reagent and Geneticin selection (Thermo Fisher Scientific). The positive cells were further enriched by fluorescence-assisted cell sorting (FACSCalibur, BD Biosciences). All NB2a cell cultures contain proliferating, spindle-like cells and differentiated, neuron-like cells. We have not observed any difference in Lasso or LPHN1 expression between these two types of cell in stably transfected NB2a cultures.

## Protein purification

For increased expression of Lasso or LPH constructs, the complete medium was replaced with a serum-free DMEM (for HEK23A cells) or Neurobasal-A containing supplements (for NB2a cells). Lasso-D was purified by immunoaffinity chromatography. Briefly, serum-free medium conditioned by HEK293A cells expressing Lasso-D was filtered through 0.2 µm filters and incubated with anti-FLAG M2 affinity gel overnight at 4°C. Lasso-D was then eluted with 20 mM triethylamine, neutralized with 1 M HEPES, dialyzed against PBS, sterile-filtered for use in cell culture and concentrated on sterile 30 kDa MWCO filtration units (Vivaspin, GE Lifesciences). Medium above non-transfected cells was processed in the same manner and used as a negative control. Amount and purity of concentrated

Lasso-D were assessed by SDS-PAGE and Coomassie staining. Activity was confirmed by measuring its binding to cell-surface or soluble LPHN1 constructs (*Silva et al., 2011*).

## Primary neuronal cultures

Hippocampal cultures were prepared from Sprague-Dawley E18 rat hippocampi (BrainBits UK), according to the supplier's instructions, or dissected from P0 AG148/2 mouse pups (*Adgrl1$^{-/-}$*, LPH1 KO) under sterile conditions. Hippocampi were digested with 2 mg/ml papain in Ca$^{2+}$-free Hibernate-A medium and dissociated in Hibernate-A medium with B27 supplement using fire-polished Pasteur pipettes. Cells were seeded in Neurobasal-A/B27 medium on poly-D-lysine-coated 13 mm coverslips at $5 \times 10^4$ cells/coverslip and maintained at 5% CO$_2$ and 37°C. The medium was partially replaced at least once a week.

## Electroporation of neurons

Primary hippocampal neurons were transfected using Amaxa Rat Neuron Nucleofector Kit (Lonza) as described by the manufacturer. Briefly, dissociated cells were resuspended in Rat Neuron Neucleofector Solution with Supplement, then mixed with 3 µg of pcDNA6-GFP and electroporated in Nucleofector using the G-013 program. The transfected cells were resuspended in 500 µl of a recovery medium, containing a 1:3 mixture of Hibernate-A/B27 and Ca-free Hibernate-A (BrainBits UK), and incubated at 37°C for 15 min. Cells were plated at a higher concentration to compensate for cell death. Next day, 0.8 nM Lasso-D was added to the medium (PBS was added to control medium). At 4 DIV, the cultured hippocampal cells were fixed with 4% paraformaldehyde (PFA), stained and visualized as described below in Image Analysis.

## Cultures in MAIDs

To investigate axonal responses to chemoattractant gradients, MAIDs (*Figure 5*) with 150 µm separation walls (Xona Microfluidics LLC) were prepared in accordance with the manufacturer's guidelines (*Harris et al., 2007a*; *Harris et al., 2007b*). Briefly, MAIDs were sterilized with ethanol, washed with sterile water and dried. To facilitate firm attachment of MAIDs, $22 \times 22$ mm coverslips (VWR International) were sonicated in water and ethanol, autoclaved, dried, then coated with 1 mg/ml poly-D-lysine overnight, washed, and dried overnight before the assembly.

For neuronal cell culture in MAIDs, E18 rat hippocampi were dissociated as above. Neurons ($1.5 \times 10^5$/10 µl) were added to Somal Compartments and allowed to settle for 30 min. MAIDs were then filled with Neurobasal-A/B27. After 3 DIV, the medium in Axonal Compartments was carefully replaced with medium containing soluble Lasso-D or with control medium. Alternatively, HEK293A cells stably expressing Lasso-D (or untransfected) were plated in the wells of Axonal Compartment. At 8 DIV, the cells were fixed and processed as described below.

## Protein diffusion in MAIDs

For diffusion modeling experiments, MAIDs were assembled as above and filled with PBS; then 0.1 mg/ml BSA-TRITC (Thermo Fisher Scientific) in PBS was added to Axonal Compartments without changing liquid level in any compartment (to avoid creating a hydrostatic pressure in the microchannels). BSA-TRITC diffusion in MAIDs was monitored by time-lapse fluorescent imaging of all compartments for 5 days under an Axiovert fluorescent microscope (Carl Zeiss) equipped with a temperature- and humidity-controlling enclosure, and a Canon G5 camera. Fluorescence intensity profiles across the microchannels at multiple time points were generated in ImageJ (NIMH, Bethesda; RRID:SCR_002285, RRID:SCR_003070) and normalized to the fluorescence profile of 100 ng/ml BSA-TRITC forced into the microchannels and both compartments.

## Immunocytochemistry

Cells on coverslips or inside MAIDs were fixed for 10 min with 4% PFA (for staining requiring SDS treatment to aid epitope retrieval, the fixative also included 0.1% glutaraldehyde). Cells were permeabilized with 0.1% Triton X-100 (or 1% SDS for PAL1 and dmAb staining), washed, then blocked for 1 hr with 10% goat serum in PBS and incubated with primary antibodies in blocking solution (dilutions used were: PAL1, 3 ng/ml; dmAb, 1:300; anti-NF-H, anti-myc mAb, and anti-GFP, 1:1,000; anti-V5, 1:2,000) for 1 hr at room temperature (or overnight at 4°C with PAL1 and dmAb). The coverslips

or MAIDs were then washed three times and incubated for 1 hr with secondary antibodies in blocking solution, followed by three washes. Coverslips were mounted using FluorSave mounting medium (Calbiochem), while neurons in MAIDs were imaged within 4 hr after the washes.

## Receptor patching

NB2a cells stably expressing LPH-42 were grown on poly-D-lysine-coated coverslips in DMEM, 10% fetal calf serum (PAA Laboratories) to 30–50% confluency and to test receptor clumping incubated at 0°C for 20 min in PBS with one of the three potential LPHN1 ligands: (1) 20 nM Lasso-D, (2) 2 nM Alexa Fluor 647-labeled LTX$^{N4C}$ (*Volynski et al., 2004*), or (3) rabbit anti-NTF antibodies (RL1), followed by a 20 min incubation with Alexa Fluor 546-conjugated goat anti-rabbit IgG. In control, only the fluorescent secondary antibody was added for the last 20 min. The cells were then fixed for 10 min with 4% PFA in PBS, blocked with 10% goat serum in PBS, and subsequent procedures were designed to reveal the distribution of the three components of each assay (NTF, CTF, and ligand). First, in all experiments, the V5 epitope on LPHN1 NTF was detected with a rabbit anti-V5 antibody (1 hr in blocking solution), followed by Alexa Fluor 488-conjugated goat anti-rabbit IgG and fixation. Subsequent staining depended on the ligand used: (1) Lasso-D was stained using a mouse anti-FLAG mAb and Alexa Fluor 546-conjugated goat anti-mouse IgG. For LPHN1 CTF detection, the cells were then permeabilized with 0.1% Triton X-100, incubated with a chicken anti-myc antibody, fixed, blocked, and stained with Alexa Fluor 647-conjugated anti-chicken antibody. (2) With LTX$^{N4C}$-induced patching, the cells were permeabilized, incubated with a mouse anti-myc mAb, fixed, blocked, and stained with an Alexa Fluor 546-conjugated anti-mouse IgG. (3) With RL1-induced patching (and in controls), the cells were permeabilized, incubated with the chicken anti-myc antibody, fixed, blocked, and stained with Alexa Fluor 647-conjugated anti-chicken antibody. The primary antibodies were used at 1:1000 dilution; the secondary antibodies, 1:2000; the cells were washed three times with PBS after each stage. At the end, the cells were briefly fixed, blocked, washed, and mounted using FluorSave reagent (Calbiochem, Cat. No. 345789).

## Image acquisition

Images of axons in MAIDs were acquired on an Axiovert 200M microscope (Carl Zeiss) using LD Plan-Neofluar 20x objective and Volocity-controlled camera, filters, shutter, and stage. Images were taken with a 5% overlap to facilitate stitching (Perkin-Elmer; RRID:SCR_002668). Blank images were subtracted to correct for optical artifacts. The images were stitched automatically and 'despeckled', using a 3 × 3 median filter (ImageJ). To correct for large illumination artifacts, background was subtracted in ImageJ using the 'Subtract background' plug-in, with a 100 μm window and the sliding paraboloid algorithm.

Images of immunostained cells and neurons on coverslips (other than for neurite tracing) were acquired using an upright laser-scanning confocal microscope (LSM-510, Zeiss; RRID:SCR_014344) equipped with 40x or 100x oil-immersion objectives; 488, 543, and 633 nm lasers; and 505–530, 560–615, and >650 nm emission filters. Images for neurite tracing were acquired using Axio Observer.Z1 microscope (Zeiss) equipped with Hamamatsu ORCA-Flash 4 sCMOS camera, EC Plan-Neofluar 40x objective, Colibri 2 LED illumination and appropriate filters.

## Image analysis

To correlate the polarity of LPH1 expression and growth cone turning, GFP images of growth cones and preceding axons were traced using CorelTRACE X3 (Corel, Canada). The obtained contour images were aligned along their median line, with all axons starting at the same point. The images were then flipped so that the higher LPHN1 staining was located in the right half of each growth cone. The trajectory of respective axons was then assessed: correlation was considered positive if the axon approached its cone from the right quadrant. To plot Jeffreys confidence intervals (CI) for a binomial distribution the standard formula was used: CI = $p$ + $z$*sqrt($p$*(1 $p$)/$n$), where z = 3 for confidence level CI = 0.9973.

For profiling of neurite growth within MAID Axonal Compartments, regions of interest encompassing the depth of the compartments, were selected, avoiding artefacts (e.g. antibody aggregates or HEK cell bodies). The average fluorescence was determined as a function of distance (see *Figure 5A*) from the separation wall and binned over 100 μm intervals. Background fluorescence in

the areas beyond 1200 µm from the wall (that contained no axons) was subtracted from all other fluorescence values, and the results were used for statistical analysis as described below.

For axon fasciculation measurements in MAIDs, the width of each axon/bundle was determined in pixels at 100 µm from the separation wall and converted to µm.

Neurite tracing of GFP-positive neurons was performed in ImageJ (*Schindelin et al., 2012*) using default settings in Simple Neurite Tracer plug-in (*Longair et al., 2011*). The longest neurite for each cell was used as a single independent measurement (data obtained from three independent cultures).

Analysis of the co-localization of the NTF, CTF, and respective ligands in the plasma membrane was carried out using a method previously developed and tested (*Silva et al., 2011*). Here, the confocal images were obtained near the middle of each cell (optical plane, $Z = 0.5$ µm). For consistency, the recorded images were assigned false colors according to the detected protein, irrespective of the actual fluorescence wavelength used for detection. The fluorescence profiles for each protein along the cell's perimeter were collected using ImageJ. Pearson's correlation coefficient $r$ was then calculated for the pairs of resulting profiles obtained from 4 to 7 independent experiments.

In the representative images that were used in the Figures, the contrast and brightness were enhanced in the same manner as in respective control images.

## Fluorometry

For experiments with LPHN1 KO and WT/HET cultures in MAIDs, the membranes of cell bodies and axons were labeled using 5 µM DiO (Vybrant DiO, Life Technologies) in Neurobasal-A, containing B-27 supplement and 0.005% Pluronic F-127 (Sigma-Aldrich), which had been passed through a 0.2 µm filter. After 30 min incubation, the excess dye was carefully washed two times, and the cell bodies (Somal Compartments) and axons (Axonal Compartments) were solubilized in 1% Triton X-100 in PBS. The undiluted axonal and 10-fold diluted somal fractions were analyzed in microtiter plates using a Fluoroskan Ascent Fluorometer (485 nm excitation, 505 nm emission filters) (Thermo Fisher Scientific). In some experiments, 2 µL samples of lysates were individually measured using a Nano-Drop ND-3300 Fluorospectrometer (Thermo Fisher Scientific) with the following settings: 470 nm Blue LED excitation, 500–700 nm emission spectrum, quantified at 504 nm. The levels of fluorescence were proportional to the amount of axons/cells bodies present in respective compartments.

## Western blotting

For Western Blot analysis of conditioned media, these were passed through 0.2 µm low protein-binding filters (PALL, USA). The cells on coverslips were lysed in ice-cold RIPA buffer (1% sodium deoxycholate, 0.1% SDS, 1% Triton X-100; 10 mM Tris-HCl, pH 8; 140 mM NaCl), supplemented with protease inhibitors and 1 mM EDTA. To prepare samples for electrophoresis, the cell lysates and media were incubated at 50°C for 30 min with sample buffer containing 2% SDS and 100 mM DTT. The samples were separated on standard SDS-containing polyacrylamide gels, blotted onto polyvinylidene fluoride membrane (Immobilon-P, IPVH00010, Merck), blocked with 5% non-fat dry milk, incubated with primary antibodies diluted in 2% BSA for TN2N or 5% milk for all other antibodies (dilutions used were: PAL1, 1:500; dmAb, 1:1,000; TN2N, 1 µg/ml; actinin, 1:1,500; NF-H, 1:10,000) and respective horseradish-peroxidase conjugated secondary antibodies. The stained membranes were visualized by WestFemto chemiluminescent substrate kit (Thermo Fisher Scientific) and LAS3000 gel/blot documentation system (FUJIFILM).

## Measurements of cytosolic Ca2+

Cytosolic $Ca^{2+}$ concentration was monitored using Fluo-4 $Ca^{2+}$ indicator (the method was also described in (*Silva et al., 2009*; *Volynski et al., 2004*). The stably transfected NB2a cells expressing LPH-42 were pre-incubated in serum-free medium for 24 hr in 30 mm dishes. Then the cells were equilibrated for 20 min in physiological buffer (in mM: NaCl, 145; KCl, 5.6; glucose, 5.6; $MgCl_2$, 1; EGTA, 0.2; HEPES, 15; pH 7.4; BSA, 0.5 mg/ml) containing 2.5 µM Fluo-4 acetomethoxy ester (Fluo-4-AM, Thermo Fisher Scientific) and 10% Pluronic F–127, washed and further incubated for 20 min for dye de-esterification. LPHN1-expressing cells were identified by staining with primary mouse anti-V5 mAb pre-labeled with Alexa Fluor 568 (Zenon, Thermo Fisher Scientific). Images were acquired every 5 s under the LSM510 microscope using a 40x Achroplan water-dipping objective,

488 nm laser and a 505–550 nm band-pass emission filter. The following protocols were typically applied (the addition times and final concentrations of the additives are indicated, see also *Figure 7—figure supplement 1A and B*). *Protocol 1*: 0 min, baseline recording; 5 min, 1 nM LTX$^{N4C}$, 360 nM Lasso-D, or control buffer; 30 min, 2 mM Ca$^{2+}$; 50 min, 1 nM wild-type α-LTX; 55 min, end. *Protocol 2*: 0 min, 2 mM Ca$^{2+}$, baseline recording; 5 min, 360 nM Lasso-D or control buffer; 30 min, 1 nM LTX$^{N4C}$; 80 min, 1 nM α-LTX; 90 min, end. Ca$^{2+}$ fluorescence of individual positive cells was quantified using the LSM510 software and normalized between the starting fluorescence and maximal fluorescence induced by α-LTX.

## Electrophysiology

MEPPs were recorded from isolated neuromuscular preparations by the method also used in (*Lelyanova et al., 2009*). *Flexor digitorum brevis* muscles were dfrom male P21 mice (C57BL/6J: *Adgrl1$^{+/+}$* or *Adgrl1$^{-/-}$*), cleaned from connective tissue, fixed using entomological pins in Petri dishes pre-coated with Sylgard silicone polymer (Dow Corning), and incubated in constantly oxygenated physiological buffer containing (in mM): NaCl, 137; KCl, 5; MgCl$_2$, 1; EGTA, 0.2; glucose, 5.6; HEPES, 10; pH 7.5; tetrodotoxin (Latoxan), 0.001. Sharp electrodes with tip diameter <0.5 μm and 30–60 MOhm impedance were produced on a P-97 puller (Sutter) from borosilicate glass filament capillaries (1.5 mm; World Precision Instruments) and filled with 5 M ammonium acetate. Spontaneous presynaptic activity (based on MEPPs detection) was recorded using a system consisting of an Axoclamp 2B pre-amplifier (Axon Instruments) in the current clamp mode, a secondary differential amplifier with a high-frequency filter (LPF202A, Warner Instruments), a HumBug harmonic frequency quencher (Quest Scientific), a Digidata 1322A digitizer (Axon Instruments), and a microcomputer running AxoScope software (Axon Instruments). The recorded traces were subsequently analyzed using MiniAnalysis software (Synaptosoft Inc.).

## Quantification and statistical analysis

The data shown are the means ± SEM, unless otherwise stated. A Lilliefors test was applied to all data sets to assess normality in data distribution. Statistical significance was then determined using two-tailed heteroscedastic t-test, with Bonferroni correction in cases of multiple pair-wise comparisons. For non-normally distributed data, a Mann-Whitney test was applied. The axonal fluorescence curves obtained from image analysis in MAIDs were compared using n-way ANOVA algorithm (MATLAB; RRID:SCR_001622), where n reflected the number of factors involved in an assay (treatment type, distance from the separation wall and batch number) (*Vysokov, 2018*). To test for correlation in axonal fasciculation measurements, a Pearson correlation coefficient ($R^2$) and the *p* values (to test the correlation hypothesis) were calculated using MATLAB. Jeffreys confidence intervals were used to assess statistical significance of correlation between LPH1 enrichment and growth cone turning direction. Differences were considered significant if $p < 0.05$. The specific *p* and *n* values are indicated in corresponding figure legends or the following notation is used to denote statistical significance: NS (non-significant), $p > 0.05$; *, $p < 0.05$; **, $p < 0.01$; ***, $p < 0.001$. The investigators were blinded to the identity of samples during data collection and analysis in all experiments involving LPHN1 KO.

## Acknowledgements

Supported by a Wellcome Trust Project Grant WT083199MF, a Biotechnology and Biological Science Research Council Core Support Grant BBF0083091, and core funding from the University of Kent School of Pharmacy (to YAU); and in part by the Russian Scientific Foundation Grant 14-44-00051 and 16-19-10597 (to AGT).

## Additional information

### Competing interests

Nickolai V Vysokov: affiliated with BrainPatch Ltd. and has no other competing interests to declare. John-Paul Silva: affiliated with UCB-Pharma. The author has no other competing interests to declare. Jason Suckling: affiliated with Thomsons Online Benefits. The author has no other competing interests to declare. John Cassidy: affiliated with Arix Bioscience. The author has no other competing

interests to declare. Alexander G Tonevitsky: affiliated with Scientific Research Centre Bioclinicum. The author has no other competing interests to declare. The other authors declare that no competing interests exist.

## Funding

| Funder | Grant reference number | Author |
|---|---|---|
| Wellcome Trust | WT083199M | Yuri A Ushkaryov |
| Biotechnology and Biological Sciences Research Council | BBF0083091 | Yuri A Ushkaryov |
| Russian Science Foundation | 14-44-00051 | Alexander G Tonevitsky |
| Russian Science Foundation | 16-19-10597 | Alexander G Tonevitsky |
| University of Kent | | Yuri A Ushkaryov |

The funders had no role in study design, data collection and interpretation, or the decision to submit the work for publication.

## Author contributions

Nickolai V Vysokov, Conceptualization, Data curation, Software, Formal analysis, Investigation, Visualization, Methodology, Writing—original draft, Writing—review and editing; John-Paul Silva, Formal analysis, Investigation, Visualization, Methodology; Vera G Lelianova, Jason Suckling, Formal analysis, Investigation, Methodology; John Cassidy, Formal analysis, Investigation, Visualization; Jennifer K Blackburn, Formal analysis, Investigation, Visualization, Writing—review and editing; Natalia Yankova, Formal analysis, Investigation, Writing—review and editing; Mustafa BA Djamgoz, Resources, Supervision, Project administration, Writing—review and editing; Serguei V Kozlov, Conceptualization, Resources, Investigation, Methodology, Writing—review and editing; Alexander G Tonevitsky, Resources, Funding acquisition, Writing—review and editing; Yuri A Ushkaryov, Conceptualization, Resources, Data curation, Formal analysis, Supervision, Funding acquisition, Validation, Investigation, Visualization, Writing—original draft, Project administration, Writing—review and editing

## Author ORCIDs

Nickolai V Vysokov (iD) https://orcid.org/0000-0002-1856-1426
John-Paul Silva (iD) https://orcid.org/0000-0001-7239-8554
Yuri A Ushkaryov (iD) http://orcid.org/0000-0002-5712-8297

## Ethics

Animal experimentation: This study was conducted in strict accordance with the guidance provided in the Animals (Scientific Procedures) Act 1986 from HM Home Office. The breeding of animals (mice and rats) was carried out at Charles River UK according to approved institutional animal care. A minimal number of animals was euthanized when required and only used to produce tissue or cell preparations.

## Decision letter and Author response

Decision letter https://doi.org/10.7554/eLife.37935.027
Author response https://doi.org/10.7554/eLife.37935.028

# Additional files

## Supplementary files

• Transparent reporting form
DOI: https://doi.org/10.7554/eLife.37935.025

## Data availability

Source data files have been provided for Figures 1 and 3-7. The MATLAB source code for axonal guidance analysis has been made available on GitHub (https://github.com/artificialbrain-tech/Axon-Guidance-Scripts; copy archived at https://github.com/elifesciences-publications/Axon-Guidance-Scripts).

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
