## [Decision Letter]

Thank you for submitting your article "Proteolytically released Lasso/teneurin-2 induces axonal attraction by interacting with latrophilin-1 on growth cones" for consideration by *eLife*. Your article has been reviewed by three peer reviewers, and the evaluation has been overseen by a Reviewing Editor and Marianne Bronner as the Senior Editor. The following individual involved in review of your submission has agreed to reveal his identity: Hugo J Bellen (Reviewer #1).

The reviewers have discussed the reviews with one another and the Reviewing Editor has drafted this decision to help you prepare a revised submission.

Summary:

Ushkaryov et al. show that latrophilin-1 and Lasso/teneurin-2 interact with one another at axonal growth cones and that Lasso/teneurin-2 is capable of stimulating axonal outgrowth. Binding of Lasso/teneurin-2 to latrophilin-1 induces rearrangement of latrophilin-1 fragments that triggers a signaling cascade, leading to an increase in intracellular calcium and subsequent exocytosis. These downstream effects likely mediate the chemoattractant effect of Lasso/teneurin-2 on axons. Ushkaryov et al. address an intriguing question that is explored using sophisticated assays to investigate characteristics of axonal growth and guidance. The authors use known signaling mechanisms of latrophilin-1 to guide their experimental design and nicely characterize the role of the novel interacting partner, Lasso/teneurin-2 within this signaling cascade. Overall, the paper is strong and illustrates a mechanism for Lasso/tenuerin-2 mediated axonal outgrowth but additional studies would further strengthen the work.

Essential revisions:

1) Figure 2D, panel 2, Results (subsection “Soluble Lasso binds to cell-surface LPHN1”, first paragraph) – Lasso-Lasso interactions are only illustrated for Lasso-A with Lasso-D. Are there Lasso-Lasso interactions between Lasso-A and Lasso-A as well as between Lasso-D and Lasso-D?

2) Figure 2C, Results (subsection “Soluble Lasso binds to cell-surface LPHN1”, first paragraph) – The claim that binding of Lasso to latrophilin-1 alters latrophilin-1 is not evident in the presented images. Better representative images that demonstrate this reported difference are required or quantification of the difference in concentration of patches should be provided. This difference in latrophilin-1 staining, following Lasso binding is more apparent in Figure 6B-C.

3) Figure 4E – It is not clear why the authors combined WT and heterozygous mice to report data. Clarify why these two genetically different groups were presented together and report data for homozygous WT mice and heterozygous mice, separately.

4) Figure 7D – Authors neglect to show heterozygous and WT mice separately, so it is unclear if a phenotype is present in the heterozygous mice. Report homozygous WT as additional control for data interpretation.

5) Figure 2—figure supplement 1 – There is a difference in cell morphology in experiments with NB2a cells. Is this mutation-dependent, due to a difference in cell lines, or density of cells plated? Clarify the cause of this difference in cell morphology. When comparing staining between groups, select representative images with similar cell morphology.

---

## [Author Response]

Essential revisions:1) Figure 2D, panel 2, Results (subsection “Soluble Lasso binds to cell-surface LPHN1”, first paragraph) – Lasso-Lasso interactions are only illustrated for Lasso-A with Lasso-D. Are there Lasso-Lasso interactions between Lasso-A and Lasso-A as well as between Lasso-D and Lasso-D?

In these experiments we studied the interaction of the released Lasso fragment with the cell-surface Lasso. Structurally, Lasso-D is indistinguishable from Lasso-A within the domains involved in LPHN1 binding, so we used Lasso-D as a model of Lasso-A. As shown in Figure 2, we did not see any interaction of Lasso-D with cell-surface Lasso-A in our conditions.

We also conducted some experiments with the medium containing released Lasso-A fragment and cells expressing Lasso-FS. Lasso-FS has exactly the same extracellular structure as Lasso-A, but lacks the C-terminal tag. Again, we did not see any discernible interaction of Lasso-A with cell-surface Lasso. This result is now included in a new Figure 2D, panel 3.

These data are indirectly supported by Boucard et al., 2014 and Li et al., 2018, who do not see homophilic cell adhesion between Ten-2 or Lasso-A expressing cells.

In contrast, in the same paper, Boucard et al. did observe Lasso-B (ECD of Lasso-FS) binding to Lasso-A (both are gifts from our lab). However, this binding was only seen after an overnight incubation of cells with soluble Lasso, and a long incubation of dimeric proteins with dimeric targets would lead to cell surface rearrangements, creating multi-molecular complexes, which could include not only cell-surface Lasso, but also other proteins. In addition, the washes were conducted after cross-linking, i.e. under non-equilibrium conditions, so any weak and short-lived interactions were artificially preserved, likely leading to overestimation of affinity. Indeed, electron microscopy of TEN2 ECD (Lasso-B) or ECRdelta1 does not demonstrate any appreciable tetramer formation (Li et al., 2018). Moreover, size-exclusion chromatography on Superdex 200, 10/300 GL (Li et al., 2018) indicates that the major form of Lasso-A is a dimer (~550 kDa) rather than a tetramer (>1000 kDa).

Thus, if there is any Lasso-Lasso interaction, it is weak and probably unable to persist under the usual conditions of binding protocols.

2) Figure 2C, Results (subsection “Soluble Lasso binds to cell-surface LPHN1”, first paragraph) – The claim that binding of Lasso to latrophilin-1 alters latrophilin-1 is not evident in the presented images. Better representative images that demonstrate this reported difference are required or quantification of the difference in concentration of patches should be provided. This difference in latrophilin-1 staining, following Lasso binding is more apparent in Figure 6B-C.

The effect of the soluble fragment of Lasso on redistribution of LPH1 on the membrane was quantified in Figure 6F, where the images were also taken at higher magnification. When the image in Figure 2C, panel 2 is blown up, it also shows a characteristic patchy LPHN1 distribution caused by Lasso-D.

The impression that Lasso does not always redistribute LPHN1 on the cell surface is possibly based on Figure 2E, panel 2. The exposure, brightness and contrast of the image (in line with the methods described) were such as to show clearly the lack of LPHN1 on the Lasso-expressing cell (arrowhead) and the binding of Lasso to the LPHN1 expressing cell (arrow), but not to control cells. When the contrast is decreased (new Figure 2E, panel 2), it is now obvious to see that LPHN1 is unevenly distributed on the cell surface, in line with our claim and the images in Figure 6.

Generally speaking, Lasso-induced patching of LPHN1 is particularly apparent in high magnification images, but we thought that including zoom-ins would unnecessarily complicate Figure 2, which only highlights the fact of interaction between Lasso and LPHN1. In addition, we now also provide an additional panel (Figure 2C, panel 3), which even at a low magnification shows LPHN1 patching on the cell surface by soluble Lasso-A.

3) Figure 4E – It is not clear why the authors combined WT and heterozygous mice to report data. Clarify why these two genetically different groups were presented together and report data for homozygous WT mice and heterozygous mice, separately.

Our electrophysiological and neuronal cell culture experiments demonstrate that heterozygous and WT mice show equal responses to LPHN1 agonists (e.g. αlatrotoxin and the anti-LPHN1 scFv A1) and to depolarizing stimuli (KCl). In all experiments described here, heterozygous mice also did not display a phenotype statistically different from that of WT mice. Therefore, for statistical purposes and in this experiment only, we initially counted WT and HET mice together. However, since then we carried out experiments with WT mice, and have now included these data in the new Figures 4 and Figure 4—figure supplement 1. In this experiment, we do not have a sufficient number of data for HET neurons and only show KO and WT data.

4) Figure 7D – Authors neglect to show heterozygous and WT mice separately, so it is unclear if a phenotype is present in the heterozygous mice. Report homozygous WT as additional control for data interpretation.

We agree that this is an important question. However, it requires a more detailed analysis and should be addressed in a separate paper. Here, we didn’t want to complicate the message and only showed a difference between LPHN1-lacking and LPHN1-expressing neurons. We compared KO to HET samples because, given the similarity of WT and HET phenotypes, if the difference between KO and HET neurons is statistically significant, that would provide unequivocal evidence for the critical role of LPHN1. In other words, if statistical significance can be achieved when comparing neurons lacking LPHN1 (KO) with neurons expressing some LPHN1 (HET), then neurons expressing the maximal amount of LPHN1 (WT) would show a similar or better difference. However, we agree that not showing the WT data may be confusing. Therefore, we now include WT and KO data in the new Figure 7 (as well as in Figures 4 and Figure 4—figure supplement 1).

5) Figure 2—figure supplement 1 – There is a difference in cell morphology in experiments with NB2a cells. Is this mutation-dependent, due to a difference in cell lines, or density of cells plated? Clarify the cause of this difference in cell morphology. When comparing staining between groups, select representative images with similar cell morphology.

Neuroblastoma NB2a cultures show two types of cell morphology: spindle-like proliferating cells and neuron-like rounded cells. The former with time differentiate into the latter. However, there is no difference in LPHN1 expression between these cell types. The cells shown in Figure 2—figure supplement 1 were selected to demonstrate both transfected and WT cells with no regard to cell morphology. We have now included a statement in the paper to reflect this.